# Synaptic disruption and CREB-regulated transcription are restored by K$^+$ channel blockers in ALS

Alberto Catanese[1,*,†] (iD), Sandeep Rajkumar[1] (iD), Daniel Sommer[1], Dennis Freisem[1], Alexander Wirth[1], Amr Aly[1], David Massa-López[2], Andrea Olivieri[1], Federica Torelli[1], Valentin Ioannidis[1], Joanna Lipecka[3], Ida Chiara Guerrera[3], Daniel Zytnicki[4] (iD), Albert Ludolph[2,5], Edor Kabashi[6] (iD), Medhanie A Mulaw[7], Francesco Roselli[1,2,5,**,†] (iD) & Tobias M Böckers[1,2,***,†] (iD)

## Abstract

**Amyotrophic lateral sclerosis (ALS) is a fatal neurodegenerative disease, which is still missing effective therapeutic strategies. Although manipulation of neuronal excitability has been tested in murine and human ALS models, it is still under debate whether neuronal activity might represent a valid target for efficient therapies. In this study, we exploited a combination of transcriptomics, proteomics, optogenetics and pharmacological approaches to investigate the activity-related pathological features of iPSC-derived C9orf72-mutant motoneurons (MN). We found that human ALS$^{C9orf72}$ MN are characterized by accumulation of aberrant aggresomes, reduced expression of synaptic genes, loss of synaptic contacts and a dynamic "malactivation" of the transcription factor CREB. A similar phenotype was also found in TBK1-mutant MN and upon overexpression of poly(GA) aggregates in primary neurons, indicating a strong convergence of pathological phenotypes on synaptic dysregulation. Notably, these alterations, along with neuronal survival, could be rescued by treating ALS-related neurons with the K$^+$ channel blockers Apamin and XE991, which, respectively, target the SK and the Kv7 channels. Thus, our study shows that restoring the activity-dependent transcriptional programme and synaptic composition exerts a neuroprotective effect on ALS disease progression.**

**Keywords** ALS; CREB; hiPSC; motoneuron; synapse
**Subject Category** Neuroscience

## Introduction

Amyotrophic lateral sclerosis (ALS) is characterized by the progressive dysfunction and loss of spinal motoneurons (MN), leading to motor disability and death within 5 years of diagnosis (Hardiman *et al*, 2017). Synaptic and circuit excitability defects are typical features observed in ALS: murine models expressing mutant SOD1 or FUS have revealed that, contrary to the expectations born out of the excitotoxicity theory, only resistant MN display increased excitability in juvenile animals (Leroy *et al*, 2014), whereas vulnerable MN show reduced intrinsic excitability in pre-symptomatic adult mice (Delestrée *et al*, 2014; Martínez-Silva *et al*, 2018). Furthermore, ALS MN also display structural and functional signs of decreased excitatory synaptic inputs (Bączyk *et al*, 2020), implying a convergence of disturbances on MN recruitment.

A similar phenotype has been reported, albeit inconsistently (Wainger *et al*, 2014), in iPSC-derived MN from ALS patients. Reduced excitability, following a transient period of hyperexcitability, has been observed in *TARDBP-* and *C9orf72*-mutant MN (Sareen *et al*, 2013; Devlin *et al*, 2015; Zhao *et al*, 2020), as well as in those carrying mutations in the *FUS* or *SOD1* genes (Naujock *et al*, 2016). Nevertheless, the precise mechanisms causing MN hypoexcitability remain poorly understood in ALS animal models, as well as in human MN.

Despite the evidences on altered excitability being related to vulnerability and synaptic dysfunction, it remains unclear whether and how this disease manifestation relates to the many biological abnormalities previously described, such as autophagy overload, aggregate accumulation and ER stress (Hardiman *et al*, 2017). The

1 Institute of Anatomy and Cell Biology, Ulm University School of Medicine, Ulm, Germany
2 Deutsches Zentrum für Neurodegenerative Erkrankungen (DZNE), Ulm site, Ulm, Germany
3 Proteomics platform Necker, INSERM US24/CNRS UMS3633, Université de Paris – Structure Fédérative de Recherche Necker, Paris, France
4 SPPIN - Saints-Pères Paris Institute for the Neurosciences, CNRS, Université de Paris, Paris, Paris
5 Department of Neurology, Ulm University School of Medicine, Ulm, Germany
6 Institute of Translational Research for Neurological Disorders, INSERM UMR 1163, Imagine Institute, Paris, France
7 Internal Medicine I and Institute of Molecular Medicine and Stem Cell Aging, Medical Faculty, University Hospital Ulm, University of Ulm University, Ulm, Germany
*Corresponding author. Tel: +49 731 50023213; E-mail: alberto.catanese@uni-ulm.de
**Corresponding author. Tel: +49 731 50063147; E-mail: francesco.roselli@uni-ulm.de
***Corresponding author. Tel: +49 731 50023221; E-mail: tobias.boeckers@uni-ulm.de
†These authors contributed equally to this work as senior authors

broad-spectrum $K^+$ channel blocker 4-aminopyridine (4-AP) was employed to provide proof-of-principle evidence that interventions at the level of $K^+$ channels are sufficient to enhance MN excitability and improve MN survival (Naujock et al, 2016) and have been shown to improve electrophysiological properties, synaptic composition and survival in a mouse model of SMA as well (Simon et al, 2021). These findings strengthened the hypothesis that increasing MN firing by inhibiting $K^+$ channels, which control fundamental currents involved in MN activity such as the A (Leroy et al, 2015) and M current (Alaburda et al, 2002), might be neuroprotective even in different MN diseases. In addition, since beneficial effects have been reported when ALS MN were treated with the Kv7.2/7.3 opener retigabine (Wainger et al, 2014), the implication of $K^+$ conductances in pathogenic pathways may vary depending on the MN maturation stage. All in all, the impact of $K^+$ conductance manipulation on the broad pathobiochemistry of ALS MN, and the involved mechanisms, remains poorly characterized.

In the present study, we aimed at identifying (using ALS patients' iPSC-derived MN) activity-dependent transcriptional programmes disrupted in ALS and affecting synaptic stability, and how to restore these by targeting $K^+$ conductances which affect MN intrinsic excitability.

## Results and Discussion

We considered hiPSC-derived MN from 3 patients carrying the $G_4C_2$ hexanucleotide expansion in the C9orf72 gene (cumulatively referred to as ALS[C9orf72]) and 3 control lines: two obtained from healthy individuals and one in which the pathogenic C9orf72 mutation has been corrected with CRISPR technology (Corrected[C9orf72]) (collectively referred to as Healthy, Fig EV1A). After 21 days in vitro (DIV), cultures from both genotypes were composed of 50% neurons (84% of which were CHAT-positive), 3% GFAP-positive astrocytes and 15% Olig2-positive cells (which might represent oligodendrocyte or MN precursors; Sagner et al, 2018), whereas we did not detect the presence of mature oligodendrocytes (using the specific marker CC1 – Bin et al, 2016). In addition, we could not detect any signal of myelin basic protein (MBP) by performing Western blot analysis with the total lysate of Healthy and ALS[C9orf72] cultures, confirming the absence of oligodendrocytes in our in vitro system (Appendix Fig S1A–D). This indicated a high enrichment of spinal MN at the early stages of differentiation, unaffected by pathogenic mutations. Already at DIV 21, ALS[C9orf72] MN displayed typical ALS hallmarks such as the accumulation of toxic RNA foci (Fig EV1B) and the presence of aberrant perinuclear aggresomes highly enriched in SQSTM1/p62 (Fig EV1C and D). In addition, the decreased levels of lipidated LC3 (Fig EV1E) further confirmed a concomitant early-stage autophagy impairment (Sellier et al, 2016). TEM analysis revealed that the aggresomes of ALS[C9orf72] MN had striking morphological similarities to those previously observed in ALS[TBK1] MN at DIV 14 (Catanese et al, 2019; Fig EV1F). Interestingly, these aberrant inclusions were also enriched in the proteasome subunit 20S alpha in MN from both ALS-related genotypes (Guo et al, 2018; Fig EV1G), highlighting a dramatic impairment of both arms of the protein quality control: autophagy and proteasome.

To identify molecular mechanisms involved in excitability changes and synaptic disturbances, we contrasted the transcriptomes of MN from two ALS[C9orf72] patients with those of one healthy control and the Corrected[C9orf72] line at DIV 35 (Fig 1A). Enrichment analysis revealed that the genes significantly down-regulated in ALS[C9orf72] were involved in the maintenance of synaptic transmission and structure (Fig 1B), indicating a globally disrupted transcriptional programme involved in excitability and synaptic stability. On the other hand, the majority of up-regulated pathways were involved in stress response, DNA damage and cell death (Fig 1C). Reactome analysis confirmed a core cluster of up-regulated apoptotic processes in ALS[C9orf72] MN, whereas synaptic signalling was significantly down-regulated (Fig 1D). To further investigate the relevance of synaptic disturbances in ALS[C9orf72] pathology, we performed Gene Set Enrichment Analysis (GSEA) to compare our data to published datasets focusing on synapse biology (Appendix Fig S2A). This approach not only confirmed the down-regulation of synaptic transcripts highlighted by the complete GO biological processes analysis (Appendix Fig S2B), but also correlated the pathologic up-regulations observed in ALS[C9orf72] MN with gene classes associated with cerebral ageing and stress (Appendix Fig S2C; Ho et al, 2016). The strong congruency of both analysis approaches confirmed previous findings showing reduced expression of synaptic genes in C9orf72-mutant cultures (Sareen et al, 2013; Selvaraj et al, 2018; Mehta et al, 2021). Thus, our data demonstrate that alterations in the synapse-related transcriptome represent an important pathologic manifestation of ALS[C9orf72] MN. On this basis, we aimed at confirming and elucidating the contribution of synaptic aberrations to MN degeneration.

First, the PASTAA algorithm (which identifies the transcription factors associated with a specific set of genes; Roider et al, 2009) recognized a significant association of the activity-dependent transcription factor CREB (which regulates synaptic stability and synaptogenesis Lesiak et al, 2013; Dhar et al, 2014) with the altered gene expression in ALS[C9orf72] MN (Fig 1E). Indeed, we confirmed the down-regulation of the synaptic genes NLGN3, SLC6A, TRIM9, SYT1, SYNGR1, and SNAP91 in all the ALS[C9orf72] lines by single-tube qPCR (Fig 1F). This suggested that CREB transcriptional dysregulation may importantly contribute to the loss of synaptic transcripts and MN vulnerability in ALS[C9orf72]. We further investigated the phosphorylation of S133 (i.e. activation) of CREB itself (pCREB[S133]) over time. Significantly higher levels of nuclear pCREB[S133] were found in ALS[C9orf72] MN at DIV 21 (when cytosolic aggregates were first detected) compared to healthy controls. Interestingly, pCREB[S133] levels displayed a substantial progressive reduction in the mutant cultures: they were in fact lower than their DIV 21 baseline after 3 additional weeks, and at DIV 56, they significantly dropped below the levels of Healthy MN (Fig 1G). Importantly, the decline of pCREB[S133] in ALS[C9orf72] cultures started before any sign of neuronal loss. The number of CHAT-positive cells still remained comparable between genotypes until DIV56, and the MN loss became evident at DIV 70 in all the 3 ALS[C9orf72] lines (Fig 1H). Thus, ALS[C9orf72] MN display an increased phosphorylation of CREB at the early maturation stages, matching the intrinsic hyperexcitability described at the early stages of culture. The subsequent decline of pCREB[S133] follows similar dynamics as those of activity loss observed over time (Devlin et al, 2015) and, together with the reduced transcription of synaptic genes, appears to significantly contribute to the degenerative processes in ALS[C9orf72] cultures. The down-regulation of CREB phosphorylation was not the only

abnormality affecting the CREB pathway: in fact, we found that CBP, a transcriptional co-factor essential for the efficient transcription of synaptic genes (Wood *et al*, 2006), was sequestered within the aberrant aggresomes of ALS[C9orf72] MN (Fig 1I). This suggests that CREB transcriptional programmes are the target of converging pathobiochemistry in ALS[C9orf72], affecting CREB itself and its co-factors, which ultimately result in profound synaptic phenotypes and MN vulnerability.

We then assessed whether ALS[C9orf72] MN would also display abnormalities in the number, structure and distribution of synapses, as suggested by the alteration in the transcription of synaptic genes. We found that the density of excitatory synaptic contacts (defined as colocalization between the pre-synaptic marker Bassoon and the post-synaptic scaffold protein Shank2) was significantly reduced in mutant MN (Fig EV2A). Interestingly, the remaining Shank2+ post-synaptic clusters of ALS[C9orf72] MN were significantly larger than their Healthy counterparts (Fig EV2B), while the size of Bassoon[+] terminals was comparable between genotypes (Fig EV2C). Nevertheless, the global levels of the synaptic vesicle protein synaptophysin substantially declined at DIV 70 in ALS cells (Fig EV2D), in agreement with the global loss of Bassoon[+] terminals observed in immunofluorescence and confirming a general synaptic disturbance associated with the down-regulation of CREB transcriptional programme. In agreement with this, pharmacological inhibition of CREB (by the 666-15 inhibitor – Appendix Fig S3A) was sufficient to trigger the loss of excitatory synapses and ultimately the loss of MN in Healthy cultures (Appendix Fig S3B and C).

We then asked whether the alteration of the CREB pathway and synapse numbers might represent a convergent pathomechanism shared by a broader spectrum of ALS cases. Based on the common role of both C9orf72 and TBK1 in regulating the early phases of autophagy (Sellier *et al*, 2016) and on the strongly convergent autophagic phenotype (Catanese *et al*, 2019; Fig EV1), we speculated that ALS[TBK1] and ALS[C9orf72] MN might share CREB-related synaptic alterations as well. Indeed, we found that CBP was sequestered in the aggresomes also in ALS[TBK1] MN (Fig EV3A). Moreover, TBK1-mutant cells displayed a switch from up- to down-regulated pCREB[S133] levels similar to what observed in ALS[C9orf72] cultures (Fig EV3B and C). We also found that the expression levels of the CREB-dependent post-synaptic gene *Homer1* was reduced in ALS[TBK1] (Fig EV3D), which showed a significantly reduced number of Shank2:Bassoon synapses than MN from an aged-matched healthy control (Fig EV3E).

Since ALS[C9orf72] and ALS[TBK1] MN show characteristic formation of aggresomes, we tested whether the presence of cytotoxic aggregates might be sufficient to trigger CREB-dependent synaptic alterations in neurons. To this end, we created a AAV9-poly(GA)[175]-EGFP (henceforth poly(GA)) vector based on the construct previously published by May *et al* (2014). Poly(GA) aggregates are the most abundant toxic product translated from the GGGGCC expansion (Mackenzie *et al*, 2015) and, despite their detection by immunofluorescence in hiPSC-derived neurons has been inconsistently reported, their overexpression recapitulates important C9orf72-related pathological features (May *et al*, 2014). Primary cortical neurons were transduced with AAV9-poly(GA)[175]-EGFP at an efficiency of 78% (Appendix Fig S4A), and poly(GA)[+] cortical cells accumulated poly(GA)-SQSTM1/p62+ cytoplasmic inclusions (Appendix Fig S4B) and had reduced number of primary dendrites

(Appendix Fig S4C), has originally shown by May and colleagues. Notably, poly(GA)[+] neurons showed significantly increased levels of pCREB[S133] at DIV 14 (Appendix Fig S4D), but at DIV 28, these were drastically lower than in EGFP-expressing cells (Appendix Fig S4E), revealing a dynamic alteration similar to human ALS MN.

We then tested whether aggregate accumulation might be sufficient to disrupt synaptic numbers and structure. In DIV14 poly(GA)-expressing neurons, the cluster number and size of Bassoon and Shank2 were comparable in poly(GA) and EGFP groups (Appendix Fig S5A–C). In contrast, at DIV 28 (Appendix Fig S5D) the density of Shank2:Bassoon synapses was significantly lower in poly(GA) neurons (Appendix Fig S5E). Moreover, poly(GA) accumulation increased the size of Shank2 and Bassoon clusters (Appendix Fig S5F and G). Thus, the synaptic phenotype induced by poly(GA) closely recapitulates the one observed in hiPSC-derived ALS MN and in *post-mortem* tissue (Henstridge *et al*, 2017).

These results indicate that the dynamic dysregulation of pCREB[S133] and synapse loss are pathological features shared by different ALS[C9orf72] models. In this context, the role played by the accumulation of cytotoxic aggregates appears to be crucial. In fact, these pathological features can be triggered by acutely expressing poly(GA) in primary neurons, and impaired synaptic transcriptome has been found also when overexpressing poly(PR) as well (Maor-Nof *et al*, 2021). The accumulation of such aberrant structures has indeed been associated with increased ER and oxidative stress (Zhang *et al*, 2014), which alter CREB activation (Seo *et al*, 2010; Pregi *et al*, 2017). Notably, we did not detect GA aggregates in ALS[C9orf72] MN, and the poly(GA) overexpression did not induce CBP aggregation. Nevertheless, Vasp, which interacts with CREB (Tateya *et al*, 2019) and controls synaptic dynamics (Lin *et al*, 2010), has been found to be highly enriched within GA aggregates (Radwan *et al*, 2020). Thus, these alterations seem to be triggered by the accumulation of aggregates, independently from their specific nature and composition. Accordingly, these pathologic features are shared by different forms of neurodegeneration, characterized by heterogeneous underlying pathomechanisms and mutations. We have in fact shown (here and previously – Catanese *et al*, 2019) that ALS[TBK1] MN are characterized by a very similar phenotype. In addition, accumulation of cytotoxic polyQ induces cell death while sequestrating CBP (Jiang *et al*, 2003), while reduced levels of pCREB have been detected in Alzheimer's *post-mortem* tissue and in a mouse model of Huntington's disease (Sugars *et al*, 2004; Bartolotti *et al*, 2016). This raises the intriguing possibility that CREB dysregulation might represent a convergent and crucial pathologic feature of different neurodegenerative conditions. Based on these considerations, we reasoned that rescuing CREB activity might have a beneficial impact on ALS-related MN.

Since CREB phosphorylation is driven by neuronal activity (Deisseroth *et al*, 1998; Wu and Deisseroth, 2001), we reasoned that enhancing neuronal firing might be sufficient to ameliorate disease markers. As proof of concept, we first used *in vitro* optogenetics to stimulate poly(GA):channelrhodopsin-expressing cortical neurons at 1Hz for 4 consecutive days. We readily verified that this stimulation protocol significantly reduced the accumulation of poly(GA)-EGFP aggregates (Appendix Fig S6A) and increased the intensity of pCREB[S133] (Appendix Fig S6B). In contrast, high-frequency stimulation (10 Hz) increased the load of poly(GA)-EGFP aggregates in transduced neurons (in agreement with Westergard *et al*, 2019;

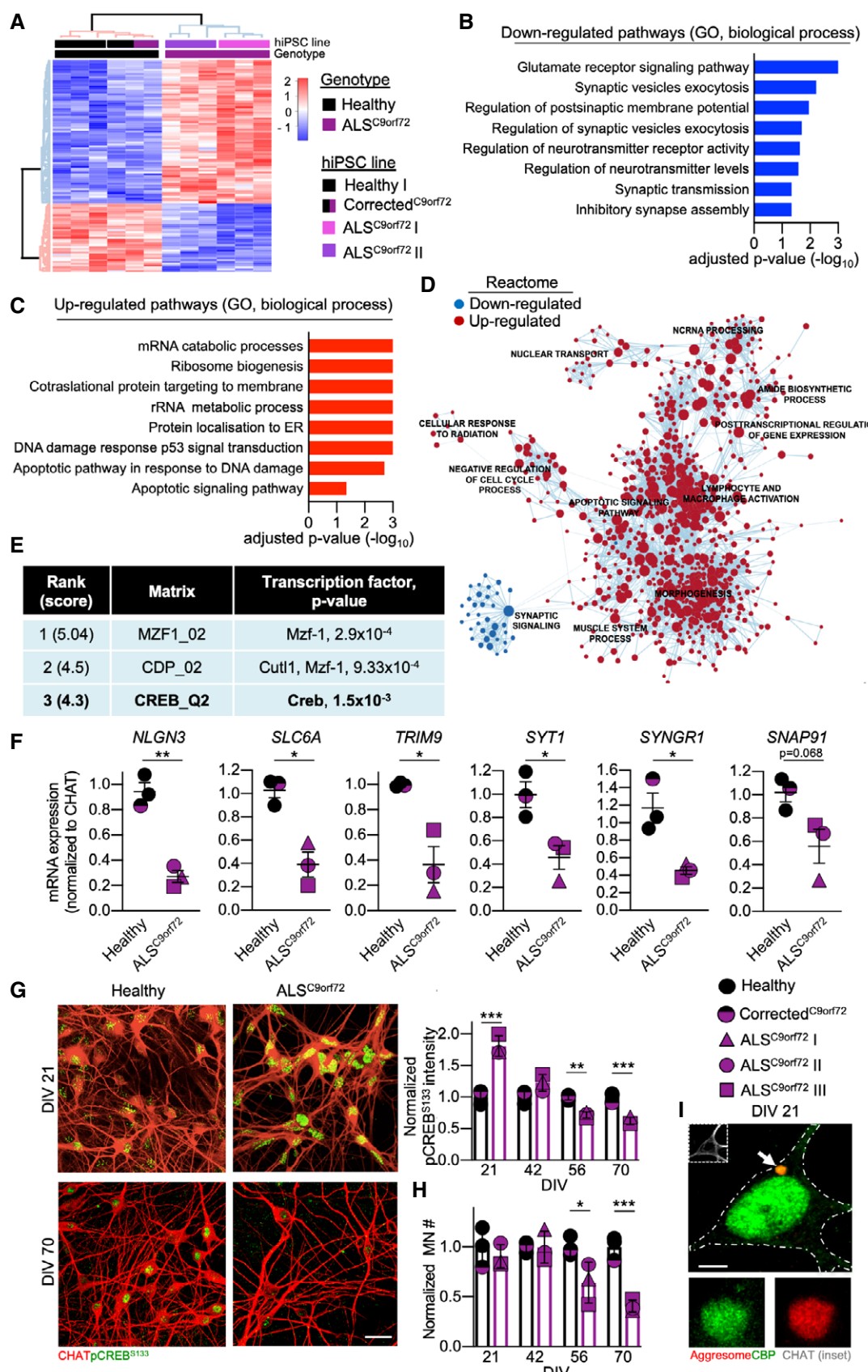

**Figure 1.**

**Figure 1. Altered transcriptional programme and CREB activation in ALS.**

A   Heatmap representing the differentially expressed genes in ALS[C9orf72] (ALS[C9orf72] I and ALS[C9orf72] II lines) *us* Healthy (Healthy I and Corrected[C9orf72] lines) MN transcriptomes. Colour scale represents gene expression relative to the Healthy genotype.

B, C   GO (biological processes) terms of the top significant down- or up-regulated pathways in ALS[C9orf72] MN (hypergeometric test).

D   Reactome representation of clustering and distribution of the significantly up- and down-regulated pathways in ALS[C9orf72] MN.

E   PASTAA ranking of the top 3 transcription factors involved in the regulation of the differentially expressed genes in ALS[C9orf72] cultures. Significance was set with the algorithm published in Roider *et al* (2009).

F   Single tube qPCR experiments showing the down-regulation of the synaptic genes *NLGN3, SLC6A, TRIM9, SYT1, SYNGR1* and *SNAP91* in ALS[C9orf72] MN (Welch's *t*-test). *n* = 3 independent cultures for each hiPSC line.

G, H   Time course analysis of nuclear pCREB[S133] levels and MN survival in ALS[C9orf72] and Healthy cultures (two-way ANOVA). *n* = 3 independent cultures for each hiPSC line (for each time point). Scale bar: 25 μm.

I   Representative confocal image showing colocalization of the CREB interactor CBP within cytotoxic aggresomes. Scale bar: 5 μm. Dashed line represents the cell soma.

Data information: *$P < 0.05$; **$P < 0.01$; and ***$P < 0.001$. Error bars represent SEM. Arrow indicates the structure displayed at higher magnification. Exact *P*-values are reported in Appendix Table S1.

Appendix Fig S6C), while still increasing the levels of nuclear phosphorylated CREB (Appendix Fig S6D), suggesting a firing rate-dependent neuroprotective effect of neuronal activity. Interestingly, inactivation of cortical poly(GA)$^+$ neurons with archaerhodopsin (Han *et al*, 2011) increased apoptosis and the load of poly(GA) aggregates in comparison with unstimulated cultures (Appendix Fig S6E and F). Overexpression of archaerhodopsin did not exert any effect in absence of light stimulation (Appendix Fig S7).

To further assess whether activity restoration could rescue synaptic loss and reduced CREB activation, we explored, in the cost-effective poly(GA)-expressing primary neurons screening platform, whether selective blockade of K$^+$ channel subtypes was sufficient to ameliorate pathological readouts. Indeed, dysregulation of K$^+$ conductances was found to contribute to hypoexcitability in ALS (Devlin *et al*, 2015).

Neuronal cultures were treated with XE991, Apamin, Agitoxin, UK78282, Charybdotoxin and NS6180 (Appendix Fig S8A) for 7 days, each at concentrations ranging between 10 nM and 10 μM. XE991 (10 and 5 μM), Apamin (500 nM) and Charybdotoxin (500 and 100 nM) reduced the total poly(GA) burden assessed as EGFP intensity with a multiplate reader (Appendix Fig S8B), indicating these molecules as potential candidates. We in fact confirmed this beneficial effect for Apamin (500 nM) and XE991 (10 μM), which reduced the aggregate burden measured in single neurons by microscopy (Appendix Fig S8C) and decreased the levels of activated caspase 3 (Appendix Fig S8D). Since Charybdotoxin has a broader selectivity (Garcia *et al*, 1995; Nikouee *et al*, 2012), it was not investigated further.

The anti-synaptotagmin antibody feeding assay, aimed at quantifying pre-synaptic vesicle release (Catanese *et al*, 2018), confirmed that both Apamin and XE991 increased neuronal firing in treated cultures (Appendix Fig S8E). Surprisingly, we found that both K$^+$ channel blockers reduced the expression of EGFP and poly(GA)-EGFP (Appendix Fig S8F), indicating that the increased firing frequency induced by the treatments might impact the levels of these transcripts.

Notably, treatment of poly(GA)-expressing neurons with higher concentration of XE991 (100 μM) significantly reduced neuronal survival in comparison with vehicle-treated cultures (Appendix Fig S9). Thus, activity-related neuroprotection may be achieved in a dose-dependent manner, as already suggested by our optogenetic experiments.

We then investigated the effect of activity modulation in our patient-related model. First, as proof of principle, we observed that

the 1 Hz optogenetic stimulation protocol diminished the burden of aggregated SQSTM1/p62 (Fig EV4A) and elevated the levels of pCREB[S133] in ALS[C9orf72] MN (Fig EV4B). Since these results suggested that enhancing neuronal activity might be neuroprotective in patients' cells as well, we tested the effect of Apamin and XE991 in ALS[C9orf72] cultures. First, we confirmed the induction of neuronal activity upon treatment by measuring the levels of c-Fos: the immediate early gene product was barely detectable in vehicle-treated ALS[C9orf72] MN, while Apamin and XE991 drastically increased its nuclear levels (Appendix Fig S10).

We then confirmed the potential neuroprotective effect of increased activity in ALS[C9orf72] MN: both K$^+$ channel blockers increased the neurite length of MN in treated cultures (Fig 2A) and reduced the number of apoptotic cleaved caspase 3$^+$ MN (Appendix Fig S11). Furthermore, high-magnification confocal optical sections showed cytosolic aggresomes surrounded by LC3A (Fig 2B), suggesting autophagic targeting. In fact, these aberrant structures were also enriched in the autophagy-specific phosphorylated SQSTM1/p62[S403] (Matsumoto *et al*, 2011), and the blockade of K$^+$ channels decreased their size (Fig 2C), while simultaneously increasing the cytoplasmic levels of soluble SQSTM1/p62[S403] (Fig 2D). In agreement with a beneficial enhancement of catabolic pathways upon treatment, K$^+$ channel blockade significantly reduced the burden of aggregated SQSTM1/p62 (Fig 2E) in ALS[C9orf72] MN. This suggested that the reduced expression of EGFP and poly(GA)-EGFP detected in Apamin and XE991-treated primary neurons might result from increased degradation. Since neuronal depolarization regulates RNA bodies and stress granules (Pascual *et al*, 2012; Kiltschewskij and Cairns, 2020; Wong *et al*, 2021), we assessed the effect of K$^+$ channel blockade on the accumulation of GGGGCC foci by fluorescent *in situ* hybridization (FISH) in our human ALS model. We found that both molecules reduced the number of RNA foci detected in ALS[C9orf72] MN (Fig 2F), without altering the expression of *C9orf72*. Thus, the beneficial effect of increased excitability may involve mechanisms regulating mRNA as well as protein stability.

Interestingly, the levels of SK2, the target of Apamin, and KCNQ2/Kv7.2, the target of XE991, were comparable in Healthy and ALS[C9orf72] MN (Appendix Fig S12), suggesting that the currents controlled by these channels, rather than their expression, might drive the neuroprotective effect.

Finally, to explore whether Apamin and XE991 treatment resulted in a restoration of activity-dependent transcriptional

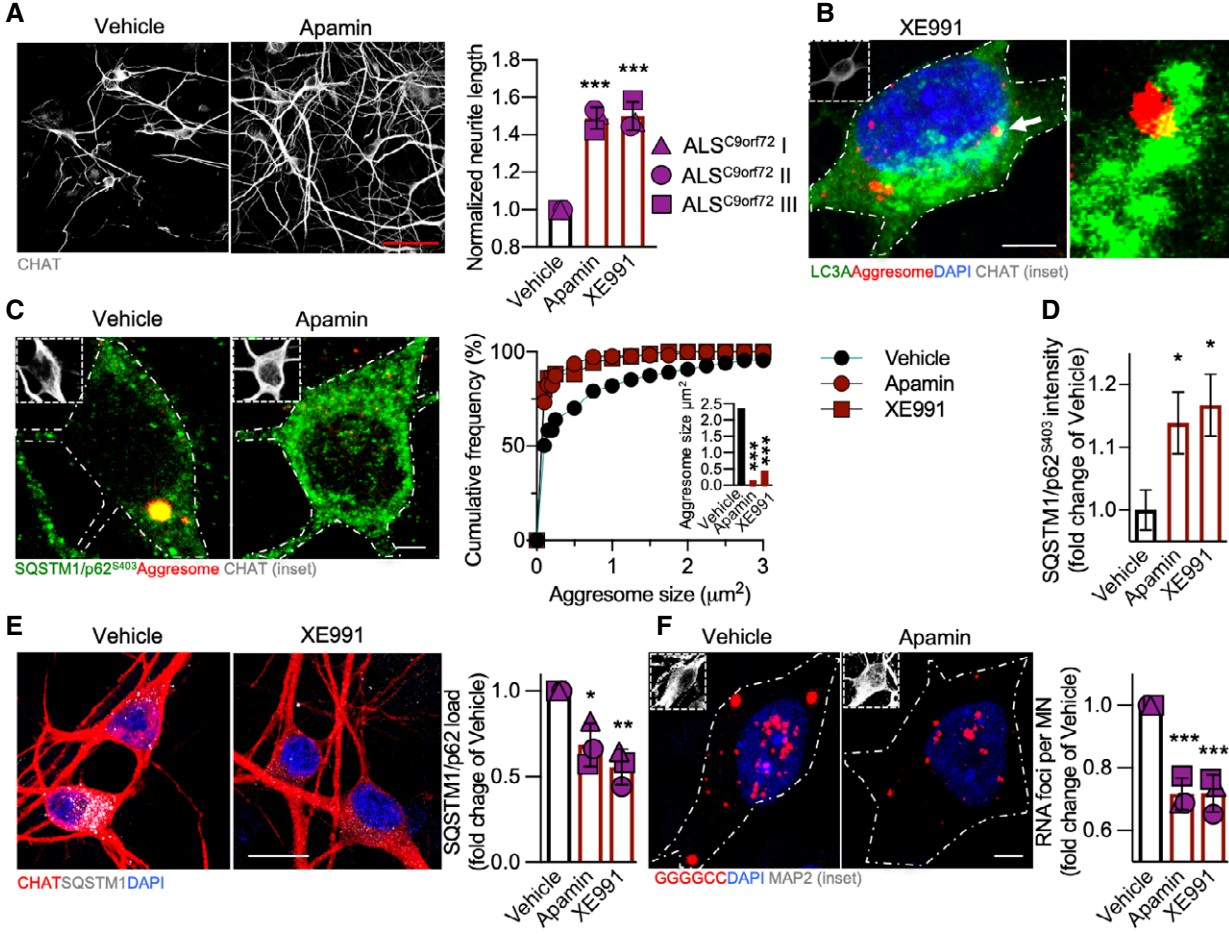

**Figure 2. Neuroprotective effect of Apamin and XE991 in ALS^C9orf72 MN.**

A   Both K$^+$ channel blockers exert a neuroprotective effect in ALS^C9orf72 cultures by increasing the neurite length of mutant MN (one-way ANOVA followed by Dunnett's multiple comparison test). $n = 3$ independent treatments with each hiPSC line. Scale bar: 50 µm.

B   Representative confocal image showing autophagic targeting of cytotoxic aggresomes in ALS^C9orf72 MN upon K$^+$ channel blockade. Scale bar: 5 µm. Dashed line represents the cell soma.

C   Apamin and XE991 decrease the size of aberrant aggresomes, confirming enhanced autophagy degradation (one-way ANOVA followed by Dunnett's multiple comparison test). $n = 3$ independent treatments with the ALS^C9orf72 I line. Scale bar: 5 µm.

D   K$^+$ channel blockers increase the levels of autophagy-specific phospho-SQSTM1/p62^S403 in ALS^C9orf72 MN (one-way ANOVA followed by Dunnett's multiple comparison test). $n = 3$ independent treatments with the ALS^C9orf72 I line.

E   The load of aggregated SQSTM1/p62 is also reduced by the treatments in ALS^C9orf72 cultures (one-way ANOVA followed by Dunnett's multiple comparison test). $n = 3$ independent treatments for each cell line. Scale bar: 20 µm.

F   Treatment with the K$^+$ channel blockers reduces the number of GGGGCC toxic RNA foci in ALS^C9orf72 MN. $n = 3$ independent treatments with each hiPSC line (one-way ANOVA followed by Dunnett's multiple comparison test). Scale bar: 5 µm. Dashed line represents the cell soma.

Data information: *$P < 0.05$; **$P < 0.01$; and ***$P < 0.001$. Error bars represent SEM. Arrow indicates the structure displayed at higher magnification. Exact $P$-values are reported in Appendix Table S1.

programmes, we compared the transcriptomes of ALS^C9orf72 MN exposed to the former or the latter K$^+$ channel blocker for 7 days starting at DIV 63 (matching the time point in which we observed a dramatic loss of synaptic contacts). Apamin treatment up-regulated the expression of 1,904 genes, while 1,945 genes were down-regulated (Fig 3A); in MN exposed to XE991, 4,932 genes were up-regulated and 2,024 transcripts were down-regulated (Fig 3B). PASTAA analysis revealed CREB as the only common transcription factor activated by Apamin and XE991 treatments (Fig 3C), and that CREB-sensitive genes represented a cluster of 263 co-regulated genes. Among them, the expression of the genes involved in

neuronal activity and synaptic structure (*FOS, FOSB, HOMER1, SLC18A2, ZYX, PPFIA1/LIPRIN* and *NTN*), as well as the expression of CREB-dependent autophagy regulators (Fig 3D), was up-regulated. Accordingly, Apamin and XE991 increased pCREB^S133 levels in cultured MN (Fig 3E).

Remarkably, treatment with Apamin or XE991 also restored the integrity of the synaptic network: blocking the K$^+$ channels in ALS^C9orf72 MN increased the number of Homer1b/c$^+$ synapses to values closer (but still lower) to those of Healthy cells, which in contrast did not show any synaptic change upon treatment (Fig 3F). Thus, interventions at the level of intrinsic excitability by targeting

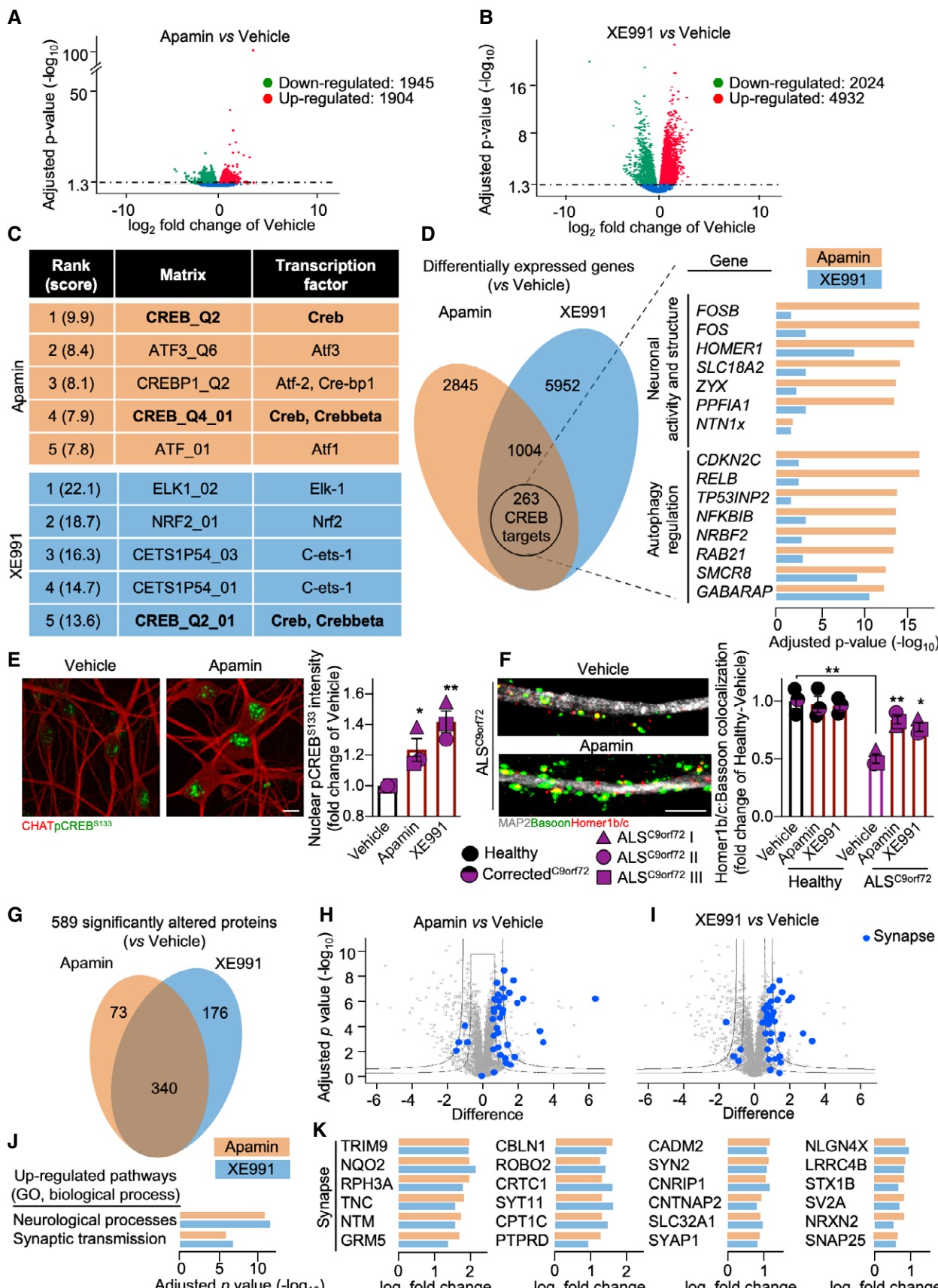

**Figure 3.**

**Figure 3.  The neuroprotective effect of the K+ channel blockers includes restoration of CREB-controlled transcription and of synapse composition.**

A, B  Volcano plots obtained by DESeq2 analysis showing the effect on the transcriptome of ALS$^{C9orf72}$ MN exerted by Apamin and XE991 in the ALS$^{C9orf72}$ II and ALS$^{C9orf72}$ III lines.

C  PASTAA ranking of the top transcription factors activated by Apamin and XE991. Significance was set with the algorithm published in Roider et al (2009).

D  Venn diagram and showing the shared 263 CREB-controlled genes significantly altered by treatment with K+ channel blockers; the bar plot represents the significantly altered genes involved in neuronal activity and autophagy obtained by DESeq2 analysis.

E  Apamin and XE991 increase the levels of pCREB$^{S133}$ in all the ALS$^{C9orf72}$ lines (one-way ANOVA followed by Dunnett's multiple comparison test). n = 3 independent treatments with each hiPSC line. Scale bar: 10 μm.

F  ALS$^{C9orf72}$ MN showed an increased number of Homer1b/c$^+$ synaptic contacts after treatment with the K+ channel blockers, which in contrast had no significant effect on Healthy MN. n = 3 independent treatments with each hiPSC line (two-way ANOVA). Scale bar: 5 μm.

G  Venn diagram showing the number of proteins significantly altered upon Apamin and XE991 treatment in ALS$^{C9orf72}$ MN. The lines ALS$^{C9orf72}$ II and ALS$^{C9orf72}$ III were used as representative of the ALS genotype.

H, I  Volcano plot displaying the up- and down-regulation of the proteins altered by Apamin and XE991 (t-test). The blue dots indicate proteins involved in synaptic structure and function.

J  Up-regulated GO (biological processes) terms in ALS$^{C9orf72}$ MN treated with either K+ channel blockers (Fisher enrichment test).

K  Synapse- and neuronal processes-related proteins significantly up-regulated by both treatments.

Data information: *P < 0.05 and **P < 0.01. Error bars represent SEM. Exact P-values are reported in Appendix Table S1.

SK (Kato et al, 2006) and Kv7 (Hu et al, 2007) channels are sufficient to restore CREB-dependent transcription and synaptic integrity.

We further assessed the impact of SK and Kv7 blockade at the proteome level, thus providing better coverage of effects on protein translation and stability. Treated MN showed a significant alteration in the levels of 589 proteins, and 340 of these changes were shared by both treatments (Fig 3G). In agreement with the restoration of synaptic networks, both K+ channel blockers caused the up-regulation of a subgroup of proteins involved in synaptic activity (referred to as "Synapse" – Figure 3H and I) and structure when compared to the vehicle group (Fig 3J). In particular, the increased levels of the synaptic vesicle-related proteins SYT11, RPH3A, SYN2, SLC32A1, SYAP1, STX1B, SV2A, SNAP25 (Rizo & Xu, 2015), and of the metabotropic glutamate receptor 5 (GRM5) showed an increased activity of the excitatory synapses in treated cultures. In addition, the up-regulation of structural proteins such as NTM, CBLN1, CADM2, CNTNAP2, NLGN4X, and NRX2 (Boeckers, 2006; Rudenko, 2019) confirmed the effect of Apamin and XE991 in re-establishing synaptic stability in ALS MN. Interestingly, the CREB interactor CRTC1, which is involved in the transcription of LTP-related genes (Zhou et al, 2006), was significantly up-regulated upon K+ channel blockade as well, confirming the role played by CREB in restoring activity-dependent transcriptional programmes in mutant MN (Fig 3K).

In line with the shared pathological features of ALS$^{C9orf72}$ and ALS$^{TBK1}$ MN, Apamin and XE991 exerted a neuroprotective effect also in TBK1-mutant cells as shown by the increased neurite length

upon treatment (Fig EV5A). Moreover, both molecules reduced the load of aggregated SQSTM1 (Fig EV5B) and of cytosolic aggresomes (although Apamin to an extent only close to significance; P-value: 0.07 – Fig EV5C). We also detected significantly higher levels of pCREB$^{S133}$ in treated ALS$^{TBK1}$ MN (Fig EV5D), together with a larger number of excitatory synapses than in vehicle-treated cells (assessed by Homer1b/c:Bassoon colocalization – Fig EV5E). Thus, the neuroprotective effect of Apamin and XE991 appears to bypass the autophagic "bottlenecks" characterizing C9orf72 and TBK1-mutant MN (Catanese et al, 2019) and to involve CREB-dependent synaptic restoration in heterogeneous ALS cases.

In conclusion, this work shows that blocking potassium channels (SK and KCNQ/Kv7), which play a crucial role in the motoneuron firing, rescues CREB-dependent transcription, restores synaptic composition and reduces accumulation of aberrant aggregates in ALS-related mutant MN. Indeed, blocking SK channels, which are responsible for the after-action potential hyperpolarizing current (AHP), has been shown to increase the firing gain and the discharge variability (Brownstone, 2006; Manuel et al, 2006) and also to contribute to the burst initiation when synaptic inhibition of motoneurons is low (Mahrous & Elbasiouny, 2017). On the other hand, blocking KCNQ channels, which are responsible for the M current, increases the intrinsic MN excitability (Alaburda et al, 2002; Lombardo & Harrington, 2016; Buskila et al, 2019). Overall, our study supports the notion that manipulation of neuronal firing might represent a valid entry point to improve neuronal fitness and delay MN loss in ALS.

# Materials and Methods

### Human iPSCs

The following hiPSC lines have been used in this study:

| hiPSC line | Gender (age) | Mutated gene | Mutation | Source |
|---|---|---|---|---|
| Healthy I | Female (45) | NA | NA | Ulm University |
| Healthy II | Male (64) | NA | NA | Ulm University |
| Corrected$^{C9orf72}$ | Male (46) | NA | NA | Cedars-Sinai (CS29iALS-C9n1.ISOnxx) |

**Table** (continued)

| hiPSC line | Gender (age) | Mutated gene | Mutation | Source |
|---|---|---|---|---|
| ALS$^{C9orf72}$ I | Male (60) | C9orf72 | $(G_4C_2)_{1.8kb}$ | Ulm University |
| ALS$^{C9orf72}$ II | Male (46) | C9orf72 | $(G_4C_2)_{6-8kb}$ | Cedars-Sinai (CS29iALS-C9nxx) |
| ALS$^{C9orf72}$ III | Female (51) | C9orf72 | $(G_4C_2)_{2.7kb}$ | Cedars-Sinai (CS30iALS-C9nxx) |
| ALS$^{TBK1}$ | Male (40) | TBK1 | p. Thr77TrpfsX4 | Ulm University |

The lines generated at Ulm University have been previously published in Catanese et al, 2019. The other lines have been commercially purchased from the iPSC Core facility of Cedars-Sinai (Los Angeles, California). hiPSCs were cultured at 37°C (5% $CO_2$, 5% $O_2$) on Matrigel®-coated (Corning, 354277) 6-well plates using mTeSR1 medium (Stem Cell Technologies, 83850). When colonies reached 80% confluence, they were detached using Dispase (Stem Cell Technologies, 07923) and passaged in 1:3 or 1:6 split ratio.

### Differentiation of hiPSC-derived MN

We differentiated MN from hiPSCs as previously described in Catanese et al (2019) (original protocol from Shimojo et al, 2015). Briefly, hiPSC colonies were detached and transferred to suspension in ultra-low attachment T25 flasks for 3 days for the formation of embryoid bodies (EBs) in hESC medium (DMEM/F12 + 20% knock-out serum replacement + 1% NEAA + 1% β-mercaptoethanol + 1% antibiotic-antimycotic + SB-431542 10 μM + Dorsomorphin 1 μM + CHIR 99021 3 μM + Purmorphamine 1 μM + Ascorbic Acid 200 ng/μl + cAMP 10 μM + 1% B27 + 0.5% N2). On the fourth day, medium was switched to MN Medium (DMEM/F12 + 24 nM sodium selenite + 16 nM progesterone + 0.08 mg/ml apotransferrin + 0.02 mg/ml insulin + 7.72 μg/ml putrescine + 1% NEAA, 1% antibiotic-antimycotic + 50 mg/ml heparin + 10 μg/ml of the neurotrophic factors BDNF, GDNF, and IGF-1, SB-431542 10 μM, Dorsomorphin 1 μM, CHIR 99021 3 μM, Purmorphamine 1 μM, Ascorbic Acid 200 ng/μl, Retinoic Acid 1 μM, cAMP 1 μM, 1% B27, 0.5% N2). After 5 further days, EBs were dissociated into single cells with Accutase (Sigma-Aldrich) and plated onto μDishes, μPlates (Ibidi) or 6-well plates (Corning) pre-coated with Growth Factor Reduced Matrigel (Corning).

### Primary rat cortical neurons

Primary cultures of rat cortical neurons were prepared from rat embryos (Sprague-Dawley rats, Janvier Laboratories) at embryonic day 18 as described previously (Catanese et al, 2018). Briefly, cerebral cortices were manually dissected under stereomicroscopic guidance and incubated for 15 min with 0.25% trypsin-EDTA (Gibco) at 37°C. The tissues were then washed once with DMEM (Gibco) (containing 10% foetal bovine serum, 1% penicillin/streptomycin and 1% GlutaMAX) and mechanically dissociated in Neurobasal Medium (Gibco) (containing 1% P/S, 1% GlutaMAX and 2% B27 – henceforth NB$^+$). After filtering through a 100-μm mesh filter, dissociated cells were then plated on poly-L-lysine-coated (Sigma-Aldrich) glass coverslips or plastic dishes and cultured in NB$^+$.

To express poly-$(GA)_{175}$-EGFP aggregates in primary neurons, we synthesized the sequence previously published by the Edbauer group (May et al, 2014) and cloned it into a pAAV backbone under the control of the human Synapsin 1 promoter to ensure neuronal expression. Viral production was done by the Penn Vector Core (University of Pennsylvania, Philadelphia, USA). Neurons were transduced at DIV 3 with either AAV9-hSyn-poly$(GA)_{175}$-EGFP or AAV9-hSyn-EGFP (a gift from Bryan Roth; Addgene viral prep # 50465-AAV9) and fixed at the time points indicated in the main text and within the figure legends.

### Optogenetics

Optogenetic experiments were performed in primary neurons, previously transduced with AAV9- hSyn-poly$(GA)_{175}$-EGFP, and transfected (using Optifect™ Transfection reagent from Thermo Fisher Scientific) with either a pAAV-CaMKIIa-hChR2(H134R)-mCherry construct (Addgene plasmid # 26975; a gift from Karl Deisseroth) to excite the cells with blue light or with a pAAV-CAG-ArchT-tdTomato construct (inhibition with yellow light) (Addgene 29778; a gift from Edward Boyden, previously described in Han et al, 2011). In hiPSC-derived MN, the experiments were carried out starting from DIV 65, and in this case, cells were transfected at DIV 63 with Lipofectamine 3000 reagent (Thermo Fisher Scientific).

Neuronal firing was manipulated with a two-colour LED array for multiwell plates (LEDA2-BY, Amuza). Optogenetic manipulations were carried out following two protocols: 1 Hz stimulation was performed by stimulating the neurons with trains of 10 pulses having a width of 500 ms and leaving a rest phase of 15 min between pulses, whereas for 10 Hz stimulation, the width of the pulses was reduced to 10 ms and 4 min of rest was allowed. After the 4 days of activity modulation, neurons were fixed, stained and imaged. To assess the cytosolic load of poly(GA) aggregates, and neuronal apoptosis (as detected by dendrite degeneration and nuclear fragmentation) in primary cells, only poly(GA)-EGFP and mCherry/tdTomato double-positive neurons were analysed. The effect of optogenetic manipulation on hiPSC-derived MN was evaluated considering only CHAT-mCherry double-positive cells. Stimulation frequencies were chosen based on previously published works investigating the firing properties of neuronal cells in SMA and ALS (Ritter et al, 2014; Wainger et al, 2014; Fletcher et al, 2017; Westergard et al, 2019).

### Pharmacological treatment

The K$^+$ channel blockers XE991, Apamin, Agitoxin, UK78282, Charybdotoxin and NS6180 (all from Tocris) were tested on primary neurons cultured in 96-well plates and transduced with AAV9-poly$(GA)_{175}$-EGFP. Treatment was carried out for 7 days by renewing half of the medium every second day. After treatment, cells were fixed and the intensity of the EGFP signal was measured with a Gen5 microplate reader (BioTek). The effect of Apamin and XE991 was confirmed in a separate set of experiments by evaluating the cytosolic

load of GA aggregates by immunofluorescence. Apamin and XE991 were also used to treat hiPSC-derived MN starting from DIV 63 until DIV 70, by replacing half of the medium every second day.

The CREB inhibitor 666-15 (Tocris) was used at a final concentration of 500 nM. The treatment was performed in hiPSC-derived MN differentiated from the Healthy I cell line, starting from DIV 63 and carried out for 24 h (when signs of neuronal stress were already visible by phase-contrast microscopy).

## Immunocytochemistry

Immunostainings were performed as previously described (Catanese *et al*, 2019). Cells were fixed with 4% paraformaldehyde (containing 10% sucrose) and incubated for two hours using blocking solution (PBS + 10% Goat Serum + 0.2% Triton X-100; the same solution was used for the incubation with primary antibodies for 24 h at 4°C). After incubation with primary antibodies, three washes with PBS were performed before incubating the cells with secondary antibodies (diluted 1:1,000 in PBS) for two hours at room temperature. Afterwards, cells were washed again three times and mounted with ProLong$^{TM}$ Gold Antifade Mountant with DAPI (Thermo Fisher Scientific) or with ibidi Mounting Medium (Ibidi).

## RNA fluorescent in situ hybridization (FISH)

Toxic RNA foci were detected in C9orf72-mutant MN cultured on μ-Slide 8-well slides (Ibidi, 80826) by performing FISH with an RNAscope$^{®}$ Multiplex Fluorescent Reagent Kit v2 (ACD-BIO, 323100) as indicated by the producer, with minor modifications to the protocol. Briefly, MN were fixed with 4% PFA for 20 min, washed twice with PBS and permeabilized for 10 min in PBS + 0.2% Triton X-100. Afterwards, cells were incubated in $H_2O_2$ for 10 min and treated with Protease III (provided with the kit and diluted 1:15 in water) for 10 min at room temperature. At this point, cells were hybridized with pre-warmed RNAscope$^{®}$ Probe GGGGCCn (ACD-BIO, 884351) in a HybEZ$^{TM}$ II Oven at 40°C for 3 h. After hybridization, cells were incubated with the amplification buffers AMP1 (for 30 min at 40°C), AMP2 (for 30 min at 40°C) and AMP3 (for 15 min at 40°C). Between each incubation step, the cells were washed twice for 5 min at room temperature with the Wash Buffer provided with the kit. Treatment with HRP-C1 reagent was performed for 15 min at 40°C before incubating the cells with the Opal 570 reagent (AKOYA BIOSCIENCES, SKU FP1488001KT; diluted 1:2,000 in TSA buffer) for 30 min at 40°C. Samples were then washed three times with Wash Buffer, incubated with HRP-Blocker reagent for 15 min at 40°C, washed again twice and processed for immunostaining as described above. For these experiments, MN were identified by using the neuronal marker MAP2. Although this is not a specific MN marker, we did not observe any specific signal by using our CHAT antibodies when performing FISH, likely due to the Protease III treatment. Nevertheless, 80% of the neurons detected in our cultures were CHAT-positive MN (as shown in Appendix Fig S1).

## Microscopy

Fluorescence microscopy was performed with an upright Axioscope 2 microscope equipped with an Axiocam 506 mono camera, and either a Plan-Neofluar 20× air (N.A. 0.5) or a Plan-Neofluar 63× (N.A. 1.25) oil immersion objective using the ZEN Blue software (Zeiss), and with a Thunder imaging system (Leica) equipped with a DFC9000 sCMOS camera, and a HC PL Fluotar 20× air (N.A. 0.4) or a HC PL Apo 63× (N.A. 1.4) oil immersion objective using the LasX software (Leica).

Confocal microscopy was performed with a laser-scanning microscope (Leica DMi8) equipped with an ACS APO 20×, 40×, or 63× oil DIC immersion objective. Images were captured using the LasX software (Leica), with a resolution of 1,024 × 1,024 pixels and a number of Z-stacks (step size of 0.3 μm for synaptic structures, otherwise 0.5 μm) enough to span the complete cell soma.

Transmission electron microscopy (TEM) was performed using a Jeol JEM 1400 (Jeol) microscope, after sample fixation by high pressure freezing (as described in Catanese *et al*, 2019).

## Image analysis

To analyse the signal intensity of specific proteins in immunostaining, the Z-stack in the channel of interest was collapsed with the maximum intensity projection tool of ImageJ, and the signal was measured within the neuronal soma (by drawing a region of interest (ROI) using the MAP2 or CHAT channel as a reference). The intensity levels of nuclear phospho-CREB were analysed by measuring the signal within a ROI overlapping with the neuronal nucleus (DAPI channel).

The neuroprotective effect of the Apamin and XE991 was assessed by analysing the neurite length using the semi-automated SNT plugin of the Neuroanatomy package for ImageJ.

In the optogenetic inhibition experiments, apoptotic neurons were recognized by degenerating neurites and fragmented nuclei within a tdTomato and poly(GA)-EGFP double-positive cell.

To analyse the neuronal cytosolic load of protein aggregates, the area occupied by these aberrant structures was calculated within a somatic ROI after thresholding. This value was then divided by the total area of the same ROI, to express the percentage of cytosol occupied by the aggregates.

To calculate the number of cleaved caspase 3$^{+}$ apoptotic cells, the signal of cleaved caspase 3 was thresholded, merged with the CHAT channel, and the double-positive cells manually counted in the entire field of view acquired.

The colocalization between different aggregate markers was evaluated within a single optical section extracted from the z-stack of the corresponding image.

To analyse the cluster size of synaptic markers, and the intensity of Syt1 dendritic puncta, three different primary dendrites for each neuron were randomly selected. A 25 μm long ROI was then drawn along each dendrite and synaptic clusters were traced with the ImageJ plugin FindFoci, using the Max Entropy algorithm.

To identify synaptic contacts, the colocalization between pre- and post-synaptic markers was quantified using Imaris software (Bitplane). First, a surface of reference was drawn in the MAP2 channel with the Surface tool. Afterwards, the puncta for each marker were detected semi-automatically in the respective channel (with the Spots tool), and the interaction between the two proteins was accepted within a minimum distance of 0.8 μm between the centre of the respective spots and with a maximum distance of 1 μm from the dendrite.

The computational parameters and post-acquisition modifications were equally applied to analyse pictures belonging to the same experiments and for figure display.

### Western blot

Western blot experiments were performed by loading 10 μg of protein (determined by Bradford Assay) in 10% acrylamide SDS–PAGE gels. After separation, proteins were transferred to a nitrocellulose membrane using a Trans-Blot Turbo device (Bio-Rad, USA). Non-specific binding sites were blocked using a 5% BSA solution (diluted in TBS pH 7.5 + 0.2% Tween 20) for two hours, and then, membranes were incubated with the primary antibody overnight at 4°C. Afterwards, blots were washed three times with TBS + 0.2% TWEEN, incubated with HRP-conjugated secondary Ab (DAKO) for two hours, and again washed. Signal was acquired using ECL detection kit (Thermo Fisher Scientific, 32106) with a MicroChemi 4.2 device (DNR Bio Imaging System). For quantification, Gel-analyser Software 2010a was used.

### Antibody list

In this study, the following primary antibodies have been used: anti-SQSTM1/p62 (Abcam, ab56416; diluted 1:500), anti-MAP2 (Synaptic Systems, 188 004; diluted 1:2,000), anti-MAP2 (Encor, CPCA-MAP2; diluted 1:2,000), anti-CREB (phospho S133) (Abcam, ab32096; diluted 1:500), anti-CBP (Cell Signaling, 7389S; diluted 1:500), anti-LC3A (Cell Signaling, 4599; diluted 1:1,000), anti-CHAT (Abcam, ab181023; a second version of the antibody was custom-made by Abcam, raised in rat; both diluted 1:500), anti-GFAP (Synaptic Systems, 173 011; diluted 1:500), anti-Olig2 (Chemicon, AB9610; diluted 1:500), anti-APC (CC1) (MERK Millipore, OP80; diluted 1:250), Proteostat® aggresome detection kit (Enzo, ENZ-51035-0025; diluted 1:5,000), anti-Proteasome 20S alpha (Abcam, ab22674; diluted 1:500), anti-Shank2 (produced by the Institute of Anatomy and Cell Biology, Ulm University; Peter *et al*, 2016), anti-Bassoon (Enzo, ADI-VAM-PS003-D; diluted 1:500), anti-Synaptophysin (Abcam, ab14692; diluted 1:1,000), anti-Synaptotagmin 1 *luminal domain* (cyanine 3-labelled) (Synaptic Systems, 105 311C3; diluted 1:500), anti-caspase 3 (Cell Signaling, 9662; diluted 1:1,000), anti-cleaved caspase 3 (Cell Signaling, 9661; diluted 1:300), anti-c-Fos (Abcam, ab208942; diluted 1:1,000), anti-Phospho SQSTM1/p62 (Ser403) (Cell Signaling, 39786S; diluted 1:500), anti-Homer1 b/c (Synaptic Systems, 160 025; diluted 1:500), anti-potassium channel SK2 (Sigma-Aldrich, P0483; diluted 1:1,000), anti-KNCQ2/$K_V$7.2 (Thermo Fisher Scientific, PA1-929; diluted 1:1,000) and anti-MBP (BioLegend, 836504; diluted 1:1,000). The secondary HRP-conjugated anti-Mouse (1:3,000 dilution) and anti-Rabbit (1:1,000 dilution) antibodies used in Western blot experiments were purchased from DAKO. For immunostainings, the following secondary antibodies from Thermo Fisher Scientific were used: goat anti-Chicken DyLight 350 (SA5-10069; diluted 1:50), goat anti-Mouse Alexa Fluor® 488 (A-11001), goat anti-Rabbit Alexa Fluor® 488 (A-11008), goat anti-Mouse Alexa Fluor® 568 (A-11004), goat anti-Rabbit Alexa Fluor® 568 (A-11011), goat anti-Chicken Alexa Fluor® 647 (A32933), goat anti-Mouse Alexa Fluor® 647 (A-21235), goat anti-Rabbit Alexa Fluor® 647 (A-21244), goat anti-Rat Alexa Fluor® 647 (A-21247) and goat anti-Guinea Pig Alexa Fluor® 647 (A-21450). All Alexa Fluor® secondary antibodies were used at 1:1,000 dilution.

### qRT–PCR

To isolate RNA from hiPSC-derived MN, the RNeasy Mini kit (Qiagen) was used following the protocol provided by the manufacturer. First-strand synthesis and quantitative real-time PCR amplification were performed in a one-step, single-tube format using the Quanti-Fast™ SYBR Green RT-PCR kit from Qiagen according to the manufacturer's instructions in a total volume of 20 μl. The primers used for qRT–PCR were purchased (Qiagen QuantiTect Primer Assays, Qiagen – validated primers without sequence information). The following settings were used: 10 min at 55°C and 5 min at 95°C, followed by 40 cycles of PCR for 5 s at 95°C for denaturation and 10 s at 60°C for annealing and elongation (one-step). The SYBR Green I reporter dye signal was measured against the internal passive reference dye (ROX) to normalize non-PCR-related fluctuations. The Rotor-Gene Q software (version 2.0.2) was used to calculate the cycle threshold values.

### mRNA sequencing

Whole-transcriptome analysis was performed at the Cambridge Sequencing Center (UK) of Novogene. A total amount of 1 μg RNA per sample was used as input material for the RNA sample preparations. Sequencing libraries were generated using NEBNext® UltraTM RNA Library Prep Kit for Illumina® (NEB, USA) following the manufacturer's recommendations, and index codes were added to attribute sequences to each sample. Briefly, mRNA was purified from total RNA using poly-T oligo-attached magnetic beads. Fragmentation was carried out using divalent cations under elevated temperature in NEBNext First Strand Synthesis Reaction Buffer (5×). First-strand cDNA was synthesized using random hexamer primer and M-MuLV Reverse Transcriptase (RNase H-). Second-strand cDNA synthesis was subsequently performed using DNA Polymerase I and RNase H. Remaining overhangs were converted into blunt ends via exonuclease/polymerase activities. After adenylation of 3' ends of DNA fragments, NEBNext Adaptor with hairpin loop structure was ligated to prepare for hybridization. In order to select cDNA fragments of preferentially 150–200 bp in length, the library fragments were purified with AMPure XP system (Beckman Coulter, Beverly, USA). Then, 3 μl USER Enzyme (NEB, USA) was used with size-selected, adaptor-ligated cDNA at 37°C for 15 min followed by 5 min at 95°C before PCR. Then, PCR was performed with Phusion High-Fidelity DNA polymerase, Universal PCR primers and Index (X) Primer. At last, PCR products were purified (AMPure XP system) and library quality was assessed on the Agilent Bioanalyser 2100 system.

The ranking of the transcription factors controlling the expression of DEGs was performed using the PASTAA tool, which identifies transcription factors that may account for transcriptional networks (Max Planck Institute for Molecular Genetics; Roider *et al*, 2009).

### NanoLC-MS/MS protein analysis

S-TrapTM micro spin column (ProtiFi, Huntington, USA) digestion was performed on 50 μg of cell lysates according to the manufacturer's instructions. Briefly, samples were reduced with 20 mM TCEP and alkylated with 50 mM CAA (chloroacetamide) for 15 min

at room temperature. Aqueous phosphoric acid was then added to a final concentration of 1.2% following by the addition of S-Trap binding buffer (90% aqueous methanol, 100 mM TEAB, pH 7.1). Mixtures were then loaded on S-Trap columns. Two extra washing steps were performed for thorough SDS elimination. Samples were digested with 2.5 μg of trypsin (Promega) at 47°C for 1 h. After elution, peptides were vacuum-dried and resuspended in 100 μl of 10% ACN, 0.1% TFA in HPLC-grade water prior to MS analysis. For each run, 1 μl was injected in a nanoRSLC-Q Exactive PLUS (RSLC Ultimate 3000) (Thermo Scientific, Waltham MA, USA). Peptides were loaded onto a μ-precolumn (Acclaim PepMap 100 C18, cartridge, 300 μm i.d. ×5 mm, 5 μm) (Thermo Scientific) and were separated on a 50 cm reversed-phase liquid chromatographic column (0.075 mm ID, Acclaim PepMap 100, C18, 2 μm) (Thermo Scientific). Chromatography solvents were (A) 0.1% formic acid in water and (B) 80% acetonitrile and 0.08% formic acid. Peptides were eluted from the column with the following gradient 5% to 40% B (120 min) and 40% to 80% (1 min). At 121 min, the gradient stayed at 80% for 5 min and, at 126 min, it returned to 5% to re-equilibrate the column for 20 min before the next injection. One blank was run between each replicate to prevent sample carryover. Peptides eluting from the column were analysed by data-dependent MS/MS, using top-10 acquisition method. Peptides were fragmented using higher-energy collisional dissociation (HCD). Briefly, the instrument settings were as follows: resolution was set to 70,000 for MS scans and 17,500 for the data-dependent MS/MS scans in order to increase speed. The MS AGC target was set to $3 \times 10^6$ counts with maximum injection time set to 200 ms, while MS/MS AGC target was set to $1 \times 10^5$ with maximum injection time set to 120 ms. The MS scan range was from 400–2,000 m/z. Dynamic exclusion was set to 30 s duration.

The MS files were processed with the MaxQuant software version 1.6.14 and searched with Andromeda search engine against the UniProt human database (release February 2020). To search parent mass and fragment ions, we set a mass deviation of 3 ppm and 20 ppm, respectively. The minimum peptide length was set to 7 amino acids and strict specificity for trypsin cleavage was required, allowing up to two missed cleavage sites. Carbamidomethylation (Cys) was set as fixed modification, whereas oxidation (Met) and N-term acetylation were set as variable modifications. Match between runs was allowed. The false discovery rates (FDRs) at the protein and peptide level were set to 1%. Scores were calculated in MaxQuant as described previously (Cox & Mann, 2008). The reverse and common contaminants hits were removed from MaxQuant output. Proteins were quantified according to the MaxQuant label-free algorithm using LFQ intensities; protein quantification was obtained using at least 2 peptides per protein.

### Data and statistical analysis

The data from pharmacological treatments and optogenetic manipulations were analysed in blinded conditions. All the experiments were performed in a minimum of $n = 3$ independent replicates (independent preparations of primary cells or independent MN differentiations from hiPSCs). No samples were excluded from the analysis, and no subjective allocation of the groups was performed. Data collection and statistical analysis were performed using Microsoft Excel, GraphPad Prism (Version 8) and RStudio software.

In the experiments with hiPSC-derived neurons, the number of cell lines used in each experiment is indicated within the figures. In those experiments where 3 hiPSC lines for each genotype were used, the average of the values obtained from all the neurons analysed from the independent differentiations (after normalization to the Healthy genotype or to the vehicle group in case of pharmacological treatment) was used to obtain a single value for each individual hiPSC line to perform statistical analysis (in order to compare the different genotypes or treatments). In those experiments where less than 3 hiPSC lines for each genotype were used, all the values from the complete dataset were used for statistical analysis.

The following statistical tests were used: to compare two independent groups, unpaired t-test with Welch correction in case of normally distributed data and nonparametric Mann–Whitney test in case of non-normal distribution, one-way ANOVA followed by the Dunnett's correction for multiple comparisons was used to evaluate differences among multiple groups if data were normally distributed, otherwise the Kruskal–Wallis test followed by Dunn's multiple comparison test was used. Two-way ANOVA was used to analyse experiments with multiple groups and multiple variables. To analyse the ratio of apoptotic neurons in inhibitory optogenetics, Fisher's exact test was used. Statistical significance was set at $P < 0.05$.

In the RNA-seq experiments, the differential expression analysis between two conditions/groups (three replicates per condition and cell line) was performed using DESeq2 R package. DESeq2 provides statistical routines for determining differential expression in digital gene expression data using a model based on the negative binomial distribution. The resulting P-values were adjusted using the Benjamini and Hochberg's approach for controlling the false discovery rate (FDR). Genes with an adjusted $P < 0.05$ found by DESeq2 were assigned as differentially expressed. Gene Ontology (GO) enrichment analysis of differentially expressed genes was implemented by the clusterProfiler R package, in which gene length bias was corrected. GO terms with corrected $P < 0.05$ were considered significantly enriched by differential expressed genes. In order to gain statistical power and reduce the variability coming from "cell line effect", the samples from the RNA-seq presented in Fig 1 were pooled according to the corresponding genotype (ALSC9orf72 and Healthy; 6 samples per group). Based on this design, differentially expressed genes and heatmaps were generated using R statistical computation environment and the Bioconductor packages limma and gplots (Huber et al, 2015; Ritchie et al, 2015; Warnes et al, 2020). Afterwards, Gene Set Enrichment Analysis (GSEA) was performed using the GSEA tool from broad MIT (Subramanian et al, 2005). The significant GO terms network was generated using Cytoscape (Shannon et al, 2003).

In NanoLC-MS/MS protein experiments, statistical and bioinformatic analysis, including heatmaps, profile plots and clustering, was performed with Perseus software (version 1.6.12.0) freely available at https://maxquant.net/perseus/. For statistical comparison, we set three groups (corresponding to vehicle, Apamin and XE991 treatments), each containing four biological replicates. Each sample was run in technical triplicates as well. We then filtered the data to keep only proteins with at least 3 valid values out 4 in at least one group. Next, the data were imputed to fill missing data points by creating a Gaussian distribution of random numbers with a standard deviation of 33% relative to the standard deviation of the measured values

**The paper explained**

**Problem**

Although different pathomechanisms have been identified and investigated in ALS, the specific contribution of synaptic alterations to the degeneration of motoneuron has not been fully clarified. In line with this, also the molecular mechanisms and co-factors involved in the synaptic disruption observed in ALS are still poorly understood. In this work, we set out to identify synapse-related alterations contributing to the loss of motoneurons, with the final aim of identifying novel neuroprotective mechanisms rescuing synaptic disturbances and delaying motoneuron degeneration in ALS.

**Results**

Human ALS-related motoneurons display a time-dependent dysregulation of CREB activity, reduced expression of synaptic genes and loss of excitatory synapses. These alterations, which contribute to the loss of motoneurons over time, can also be triggered by the acute overexpression of poly(GA) aggregates in primary neurons. This suggested a central role played by the accumulation of cytotoxic aggregates, which are indeed also found in human motoneurons harbouring pathogenic mutations in the *C9orf72* and *TBK1* genes. Notably, the $K^+$ channel blockers Apamin and XE991 rescued CREB-related transcriptomes and excitatory synapses, increased neuronal survival and reduced the accumulation of cytotoxic aggregates.

**Impact**

In this manuscript, we show that altered CREB activity and loss of synaptic contacts play a crucial role in the neurodegenerative processes linked to ALS. Our results indicate that the re-establishment of CREB-dependent transcription and synaptic contacts might represent the basis for the development of novel approaches aimed at delaying neurodegeneration in ALS.

and 1.8 standard deviation downshift of the mean to simulate the distribution of low signal values. Hierarchical clustering of all proteins was performed in Perseus on logarithmized LFQ intensities after z-score normalization of the data, using Euclidean distances. We performed a *t*-test of each treatment against vehicle, and data were plotted in a multiple volcano plot (class A proteins FDR < 0.01, S0 = 1, class B proteins FDR < 0.01, S0 = 0.1). Significant proteins class B up-regulated and down-regulated by the two treatments were submitted to Fisher enrichment test (Benjamin–Hochberg FDR < 0.02).

### Ethical approval

All procedures with human material were in accordance with the ethical committee of Ulm University (Nr.0148/2009 and 265/12) and in compliance with the guidelines of the Federal Government of Germany. All participants gave informed consent for the study. The use of human material was approved by the Declaration of Helsinki concerning Ethical Principles for Medical Research Involving Human Subjects, and experiments were performed according to the principles set out in the Department of Health and Human Services Belmont Report.

Preparation of rat primary cells was allowed by the Permit Nr. O.103 of *Land Baden-Württemberg* (Germany) and performed with respect to the guidelines for the welfare of experimental animals issued by the German Federal Government and the Max Planck Society, and the ARRIVE guidelines.

## Data availability

The datasets produced in this study are available in the following databases:

- RNA-seq: Gene Expression Omnibus (GSE168831) (https://www.ncbi.nlm.nih.gov/geo/query/acc.cgi?acc=GSE168831).
- Proteomic: ProteomeXchange (PXD020316) (http://www.ebi.ac.uk/pride/archive/projects/PXD020316).

Expanded View for this article is available online.

## Acknowledgements

The authors are thankful to Maria Manz and Sabine Seltenheim for the valuable technical support and to Prof. Paul Walther and Renate Kunz for helping with TEM analysis. The authors are also thankful to Prof. Dieter Edbauer, for providing the original poly(GA)$_{175}$-EGFP construct that was used as a reference to produce the AAV9-hSyn-poly(GA)$_{175}$-EGFP viral vector. This work was financed with funding from the Baustein Program of the Medical Faculty of Ulm University to AC (project Nr.: L.SBN.0162), and by the Deutsche Forschungsgemeinschaft (BO 1718/8-1), the Land Baden-Württemberg (Gesundheitsstandort BW, Translational Neuroscience, KNMG.001) and the Deutsches Zentrum für Neurodegenerative Erkrankungen (DZNE) to TMB. AC is also supported by the Else Kröner Fresenius Stiftung (project 2019_A111). FR is supported by the Fondation Thierry Latran (projects "Trials" and "Hypothals"), by the Radala Foundation, by the Deutsche Forschungsgemeinschaft (DFG) as part of the SFB1149 and with the individual grant no. 431995586 (RO-5004/8-1) and no. 443642953 (RO5004/9-1), by the Cellular and Molecular Mechanisms in Aging (CEMMA) Research Training Group and by BMBF (FKZ 01EW1705A, as member of the ERANET-NEURON consortium "MICRONET"). Open access funding enabled and organized by Projekt DEAL.

## Author contributions

Conceptualization: AC, DZ, EK, FR, and TMB. Investigation: AC, SR, AW, FT, JL, and ICG. Formal analysis: AC, SR, DF, VI, AW, FT, DS, AA, AO, DM-L, ICG, and MAM. Visualization: AC, and ICG. Writing – original draft: AC, DZ, and FR. Supervision: AL and TMB. Resources: AC and TMB. Funding acquisition: AC and TMB.

## Conflict of interest

The authors declare that they have no conflict of interest.

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
