## [Review Process File · EMBO Molecular Medicine]

Synaptic disruption and CREB-regulated transcription are restored by K⁺ channel blockers in ALS

Alberto Catanese, Sandeep Rajkumar, Daniel Sommer, Dennis Freisem, Alexander Wirth, Amr Aly, David Massa-López, Andrea Olivieri, Federica Torelli, Valentin Ioannidis, Joanna Lipecka, Ida Guerrero, Daniel Zytnicki, Edor Kabashi, Albert Ludolph, medhanie mulaw, Francesco Roselli, and Tobias Böckers

DOI: 10.15252/emmm.202013131

Corresponding author(s): Alberto Catanese (alberto.catanese@uni-ulm.de), Francesco Roselli (francesco.roselli@uni-ulm.de), Tobias Böckers (tobias.boeckers@uni-ulm.de)

Review Timeline:

Submission Date:	17th Jul 20
Editorial Decision:	26th Aug 20
Revision Received:	1st Apr 21
Editorial Decision:	10th May 21
Revision Received:	19th May 21
Accepted:	21st May 21

Editor: Zeljko Durdevic

Transaction Report:

Dear Dr. Catanese,

Thank you for the submission of your manuscript to EMBO Molecular Medicine. We have now received feedback from the three reviewers who agreed to evaluate your manuscript. As you will see from the reports below, the referees acknowledge the interest of the study but also raise serious concerns that should be addressed in a major revision.

Addressing the reviewers' concerns in full will be necessary for further considering the manuscript in our journal, and acceptance of the manuscript will entail a second round of review. EMBO Molecular Medicine encourages a single round of revision only and therefore, acceptance or rejection of the manuscript will depend on the completeness of your responses included in the next, final version of the manuscript. For this reason, and to save you from any frustrations in the end, I would strongly advise against returning an incomplete revision.

We realize that the current situation is exceptional on the account of the COVID-19/SARS-CoV-2 pandemic. Therefore, please let us know if you need more than three months to revise the manuscript.

I look forward to receiving your revised manuscript.

Yours sincerely,

***** Reviewer's comments *****

Referee #1 (Remarks for Author):

Amyotrophic lateral sclerosis (ALS) is an adult-onset neurodegenerative disease, in which only two FDA drugs (Riluzole and Edaravone) are used for ALS therapy. Unfortunately, these drugs could only extend the lifespan by a few months. Therefore, it is emergent to find an alternative combinatorial therapeutic approach that could significantly enhance the efficacy of ALS treatment.

In this manuscript, Catanese A et al. investigated whether neuronal excitability could be manipulated to ameliorate ALS symptoms, using the human iPSC derived motor neuron (MN) approach. Elegantly, they combined multidisciplinary approaches (i.e. transcriptomics, proteomics, optogenetics, and pharmacological treatment) to uncover that the dysregulation of CREB activity might be a hub to accelerate the hyperexcitability and motor neuron degeneration. They then tested a battery of K⁺ channel blockers (Apamin and XE991) and suggested that restoring the activity might change the CREB-dependant transcriptional program and synaptic composition. This could be a new avenue for re-examining the activity-dependent pathway for ALS drug development.

Overall, this manuscript provides two new breakthroughs: 1) CREB-dysregulated nucleocytoplasmic regulation might be a new mechanism for C9ORF72-ALS. 2) The use of specific K⁺ channel blocker cocktails could ameliorate MN hyperactivity, in some degree dependent on restoring CREB activity. However, I found it rather confusing that the "cause and consequence" of CREB dysregulation and hyperexcitability. Although the author tried to justify it might form a cross-exacerbated loop, the authors need to justify 1) whether the CREB dysregulation elicits the hyperexcitability or hyperexcitability exacerbates the CREB dysregulation; and 2) if the change of neuronal activity leads to MN degeneration, assayed by #ChAT number in the ALS-iPSC culture. A better design of the cause and consequence experiment would strengthen the conclusion of the manuscript. Besides, many of the sentences need to be reorganized to provide more precise details and concise concepts of the manuscript. Specific points are provided below:

Major points:

1. In Fig1A, what was the stage when the authors performed RNAseq in Fig1A? Did they perform the experiments before the onset of MN degeneration or after MN degeneration? This hugely influenced the conclusion if the dysregulated genes in Fig1B~C due to disease manifestation or merely a consequence of dying of MNs.
2. In many cases, the authors showed the data without controls (Fig 1H, FigEV1B, EV1E, EV1F), or no representative figures were shown to back up the statistical quantifications. It is difficult for me to evaluate the data and conclusion.
3. The application of optogenetics in Fig2 is great to elicit neuron activation, yet the readout with poly(GA) aggregation is insufficient. The quantification of MN numbers or survival upon this optogenetic manipulation, or the level of pCREB, will be more direct access to the phenotype after this manipulation.
4. The statistical method and the figure legend shall be enhanced to provide readers with correct information. For example, it not clear to me how was the survival index in FigEV1B is calculated? This could be applied to many of the supplementary figures.

Minor points:

The authors might want to revise several sentences, including lines 77~80, 83,92, 141~148, 174~176, 193~197 as I found it quite challenging to digest.

Referee #2 (Remarks for Author):

Catanese et al., investigated transcriptional profiles of human iPSC-derived motor neurons from 3 patients with mutations in C9orf72, a patient with TBK1 mutation, a patient with TBK1/FUS mutations, a mutation-corrected control of C9orf72, and two healthy controls. They found that CREB target genes were commonly altered in motor neurons from ALS patients compared to controls which was accompanied with downregulation of synapse and ion-channel associated genes. The expression levels of pCREB in motor neurons were elevated at DIV21 but decreased at DIV 70. These findings were recapitulated in rat primary cortical neurons in which EGFP-poly (GA) was overexpressed, which exhibited similar up- and down-regulation of pCREB as well as a decrease in pre-synapse numbers and morphological alteration of post-synapse. The authors also found that 1 Hz stimulation which caused channel rhodopsin-dependent activation of motor neurons decreased the amount of poly-GA but 10 Hz stimulation increased, suggesting neuronal firing was involved in C9-related ALS pathogenesis. Finally, they investigated whether selective K⁺ channel blockers to ameliorate pathological readouts. They found that both Apamin and XE-991 rescued decreased pCREB expression, altered Creb-target gene expressions, and mitigated

synaptic deficits in C9-motor neurons.

Overall, the study is well-written and the findings that K⁺-channel mediated neuronal firing is associated with vulnerability is of intrigue. However, the manuscript is still immature, and there are many issues to be addressed before publication as listed below.

Major issues

1. The data from rat primary cortical neurons were confusingly mixed with those from human iPSC-derived motor neurons. It is necessary to reconsider the order of figures. For instance, if all the data from rat cortical neurons are moved to supplementary figures, the manuscript would be more straight forward.
2. In Fig1, the authors found that 1001 from 1766 differentially expressed genes were Creb-targeted. They concluded that it was significantly higher, but they did not show how much population of Creb-targeted genes were in general.
3. All graphs must be shown with individual blots of control (vehicle) instead of dashed lines.
4. The mechanism is not clearly explained how mutations in C9orf72 and subsequent poly-GA accumulation and TBK1/FUS accumulation lead to upregulation of pCreb in nuclei followed by alteration of Creb-regulated genes.
5. Did the authors observe poly-GA accumulation in hiPSC-derived MNs with C9 mutations? If so, they can treat those motor neurons with Apamin and XE-991 to see their effects on poly-GA accumulation. If not, they need to explain the rationale to use poly-GA overexpression system in rat cortical neurons.
6. Since the authors successfully identified aggresome formation in iPSC-derived motor neurons from a case of ALS-TBK1, they can see whether Apamin and XE-991 have beneficial effects in TBK1-MNs. It would validate their conclusion that K⁺-channel mediated neuronal firing is involved in ALS pathogenesis in general.
7. In experiments of poly-GA in rat cortical neurons, the authors used AAV to overexpress EGFP-poly-GA in primary neurons. In general, AAVs are good infection efficiency to express genes or shRNAs in vivo, but not as good as lentivirus for in vitro. It is necessary to show the expression ratio of EGFP-poly-GA positive neurons.
8. In Fig2A-C, it would be more convincing if the authors chase the EGFP-poly GA-positive aggresomes of primary neurons in living before/after 1 or 10Hz stimuli.
9. To eliminate the possibility that drugs and/or optogenetic stimuli can affect transcription of EGFP-poly-GA, mRNA levels should be also measured.
10. It is necessary to confirm the neuronal activities by other methods like c-fos expression.
11. It is necessary to explain why the authors used stimulation frequencies, 1 and 10Hz for experiments.
12. In the experiment of Fig.2C, was archaerhodopsin change the amount of poly-GA accumulation?
13. The authors should describe the motor neuron purity ratio by immunostaining using specific antibodies for neurons and glial cells. For instance, the signals of cleaved PARP might be in glial cells In Fig EV4B.
14. The "survival index" used in this study is not convincing, since it represents only the ratio of MAP2 or CHAT-positive area in each image, which can be affected by morphological changes or even immunostaining or culture conditions. There are more reliable alternative indices such as neurite outlength or neurite blanch number which can be automatically measured by software.

Minor issues

1. From line 157 to 179, Fig 3B-3J must be Fig 2B-2J.
2. The data of Fig 2E-J which show the effects of candidate drugs should be separated from Fig 2A-D.
3. In Fig 1B It would be helpful if Creb-target genes are specified by different color.

4. In Fig 1E immunostaining is necessary to be shown. Similarly, it should be shown in Fig 1J as well.
5. The description of $***p<0.001$ is missing in figure legends.
6. In L770 "Scale bars: E-F 25um" must be "Scale bars: F 25um".
7. The staining images of Apamin should be included in Fig. 2G and 2I.
8. Scale bar is needed in Fig. 3F.
9. Arrows in Fig. EV1B and EV1F should be explained in legends.
10. The description of $***p<0.001$ should be omitted in the legend of Fig. EV3.

Referee #3 (Comments on Novelty/Model System for Author):

This reviewer does have a major concern regarding the initial approach to transcriptomics presented as Figure 1, as is indicated below. The inclusion of multiple ALS mutant lines in some but not all Figures is confusing, particularly since the authors have chosen to identify all as ALS-MN. For the results and conclusions drawn to have more scientific merit, it is the recommendation of this reviewer to only include C9 patient lines, indicate specifically which line produces each result within figures, and increase the number of lines utilized to have appropriate statistical power.

Referee #3 (Remarks for Author):

In this article, the authors evaluate the transcriptomics of several ALS-patient derived motor neuron lines. Initially, they identify synaptic protein dysregulation, with pCREB levels over time being a major driving factor of these differences. They screen several compounds known to modulate neuronal excitability, and identify Apamin and XE991 as capable of reducing poly-GA DPR burden in transduced primary cortical neurons. Using these compounds in ALS-MN, they then showed transcriptomic and protein profiles, along with upregulation of pCREB and synaptic components Basoon and Homer compared with vehicle conditions.

Overall, the authors propose that this work provides a framework for modulation of potassium channels in mitigating DPR burden in C9-ALS patient cells, by restoring synaptic activity and restoring autophagic processing.

This reviewer does have a major concern regarding the initial approach to transcriptomics presented as Figure 1, as is indicated below. The inclusion of multiple ALS mutant lines in some but not all Figures is confusing, particularly since the authors have chosen to identify all as ALS-MN. For the results and conclusions drawn to have more scientific merit, it is the recommendation of this reviewer to only include C9 patient lines, indicate specifically which line produces each result within figures, and increase the number of lines utilized to have appropriate statistical power.

The remaining concerns of this reviewer revolve around lack of details for clarity of interpretation.

Strengths:

The supporting evidence for autophagy impairment presented as Figure EV1 contains very striking images and robust phenotypic outcomes.

The presentation of GO analysis terms in Figure 1 C and D are well designed and organized. They allow the reader to easily come to the conclusions made by the authors in-text.

The data presented in Figure EV4 is very promising that the compounds Apamin and XE991 may be effective in increasing survival and reducing autophagic burden in C9 patient-derived neurons. Particularly, it is informative it is the functionality, rather than the expression levels of these channels that are altered.

The Materials and Methods section is extremely detailed and thorough.

Major points of concern:

A major concern for this paper lies in the initial transcriptomics evidence provided in Figures 1A and 1B. There are dramatically significant differences between the two C9 patient-derived lines presented. Furthermore the "correction" of the phenotype from line ALSC9orf72II to Corrected still has profound differences when compared with the healthy line shown. Lumping the results from only these 2 conditions in each case, and determining that is typical of C9 or "Healthy" seems to be an over representation of transcripts that are altered due to very limited biological sampling as per a power analysis justification. It is recommended that either C9 lines are compared with their own isogenic partners, or an increased line number of at least 5 per condition is required to make such conclusions.

Overall it is not clear why the TBK1 and TBK1/FUS lines were included, as the paper is focusing on the C9 phenotype, the transcriptomics and proteomics are only done with C9 comparisons, and the TBK1/FUS lines were not presented in all figures. If additional lines were needed in order to attain statistical comparison, additional C9 lines should have been utilized. Doing so will also clear up some confusion throughout the entire manuscript, as the authors refer to all lines as ALS MN, but the phenotypes and cellular pathways involved in C9, TBK1, and FUS mutations are distinct. Presently it is difficult to draw conclusions when the exact genotype is not known. Furthermore, throughout all figures it would be beneficial to somehow distinguish between the different C9 lines. This would also allow for a comparison of C9 repeat expansion lengths for the results shown. Additionally, as in Figure 3E, it would allow the reader to see if the lines respond differentially to the different compounds.

The shift in timecourse from the initial transcriptomics assessment in Figure 1 (28 DIV) to that performed in Figures 3 (DIV70) was not described or rationalized. Additionally, the specific lines that were treated and compared are not indicated, presumably comparisons were performed for the same line in vehicle and treatment conditions.

Minor points of concern:

The introduction was thorough, it would be further strengthened by inclusion of a sentence explicitly stating how manipulating K⁺ channels would alter neuronal excitability.

It is unclear why the authors chose to use primary cortical neurons rather than primary motor neurons (or healthy IPS) for their transduction/transfection experiments.

- The formation of poly(GA)-SQSTM1/p62 cytoplasmic inclusions (Fig 1H) and reduction in number of primary dendrites (Fig1I) has already been shown by May et al 2014.

- The authors state on line 162 "to further assess whether restoration of MN activity", however the experiments they then describe are in primary cortical neurons.

Figure 1J/K would be strengthened by addition of DIV14 images contrasted with DIV28 images, to show reduction of pCREB visually to the audience.

The transition from line 121 "next assessed if ALS MN would also display synaptic abnormalities" to line 122 "indeed pharmacological blockade" in healthy MN does not match up.

For Figure EV3D- were all 3 C9 and healthy lines assessed at each timepoint? It is unclear from the figure legend and presentation of the Western blot included. This is a point of concern given the spread of phenotype shown in figure EV3A. If for instance the top healthy line and TBK1 mutant were compared the effect size would be much greater than the corrected C9 and highest point C9-ALS.

The loss of synaptic components presented in FigureEV3 E-K is very interesting. However, it seems that ~20 cells/points were sampled in E and ~30 sampled for I. An equal number should be evaluated for both, particularly since there seems to be a trend at DIV14 that may be significant if additional values were assessed. This applies to all figures, as the number of cells evaluated in Figure 2A-D is also not indicated.

The results in Figure 2D are indicated as "hiPSC-derived MN" both in-text, and in-figure. Within the figure legend it says "ALS MN", which the reviewer is assuming is one of the C9 lines. It would be beneficial to explicitly state this in the text and on the figure.

It is unclear why Figure 3F does not include all 3 C9 lines. Furthermore, an evaluation of Apamin and XE991 treatments in healthy MNs is lacking. It will be meaningful to determine if the upregulations seen approach "normal" levels of synaptic components.

As there are ~250 proteins differentially altered between Apamin and XE991 treatments, would a combination be something to consider as well?

Overall, the paper would benefit from a proofreading to rectify the following small errors and potentially more that the authors will identify:

- It is standard practice for graphs to be plotted mean {plus minus} SEM with error bars in both directions. It is not indicated whether this is the case. Additionally, it would be beneficial for the reader to present these error bars with hatch marks instead of just lines.
- The "purple triangle healthy" indicated in the top of Figure 1A is not represented on the figure legend at the right hand side of that panel.
- The circles in Figure EV3B-C are too small to distinguish their colors & therefore genotypes.
- Figure 2B-H are mislabeled as C in the text.
- Figure 2F is missing indicators of statistical significance for compounds that were efficacious.

***** Reviewer's comments *****

We would like to express our gratitude to the Reviewers for their extremely constructive comments. We believe that the quality and the relevance of our manuscript strongly increased thanks to their inputs. Please find below our point-by-point rebuttal; our answers are highlighted in blue.

Referee #1 (Remarks for Author):

Amyotrophic lateral sclerosis (ALS) is an adult-onset neurodegenerative disease, in which only two FDA drugs (Riluzole and Edaravone) are used for ALS therapy. Unfortunately, these drugs could only extend the lifespan by a few months. Therefore, it is emergent to find an alternative combinatorial therapeutic approach that could significantly enhance the efficacy of ALS treatment.

In this manuscript, Catanese A et al. investigated whether neuronal excitability could be manipulated to ameliorate ALS symptoms, using the human iPSC derived motor neuron (MN) approach. Elegantly, they combined multidisciplinary approaches (i.e. transcriptomics, proteomics, optogenetics, and pharmacological treatment) to uncover that the dysregulation of CREB activity might be a hub to accelerate the hyperexcitability and motor neuron degeneration. They then tested a battery of K⁺ channel blockers (Apamin and XE991) and suggested that restoring the activity might change the CREB-dependant transcriptional program and synaptic composition. This could be a new avenue for re-examining the activity-dependent pathway for ALS drug development.

Overall, this manuscript provides two new breakthroughs: 1) CREB-dysregulated nucleocytoplasmic regulation might be a new mechanism for C9ORF72-ALS. 2) The use of specific K⁺ channel blocker cocktails could ameliorate MN hyperactivity, in some degree dependent on restoring CREB activity. However, I found it rather confusing that the "cause and consequence" of CREB dysregulation and hyperexcitability. Although the author tried to justify it might form a cross-exacerbated loop, the authors need to justify 1) whether the CREB dysregulation elicits the hyperexcitability or hyperexcitability exacerbates the CREB dysregulation; and 2) if the change of neuronal activity leads to MN degeneration, assayed by #ChAT number in the ALS-iPSC culture. A better design of the cause and consequence experiment would strengthen the conclusion of the manuscript. Besides, many of the sentences need to be reorganized to provide more precise details and concise concepts of the manuscript. Specific points are provided below:

>> Before addressing the point-by-point rebuttal (below), we would like to address the two main issues raised by the reviewer. In order to establish a causal chain of events between the reported excitability changes (hyperexcitability, then hypoexcitability) and CREB phosphorylation, we monitored the trend of the latter in C9 MN over culture maturation (shown in Figure 1). We found that the levels of pCREB¹³³ were significantly higher than those of healthy controls at DIV 21 but, were already substantially declined at DIV 42, before the appearance of any MN loss (only detectable from DIV56). These findings are in close agreement with previously published data (Wainger et al., 2014; Devlin et al., 2015; Naujok et al., 2016) showing an earlier phase of neuronal hyperexcitability (comparable to our detection of increased pCREB¹³³) in C9orf72-mutant cultures. This is followed by the progressive decline in excitability over time ultimately leading to a state of hypo-excitability (matching our detection of reduced pCREB¹³³). Thus, the decrease in CREB phosphorylation is not the consequence of the global derangement associated with imminent cell death, but rather a state that predates the cell demise by several days. These findings are in broad agreement with what was reported by Leroy et al., (2014) and Martinez-Silva et al. (2018), showing, by in vivo electrophysiology, that MN undergo a shift from hyper- to hypoexcitability in the mutant SOD1 and FUS mice. Furthermore, we show that blockade of CREB activity is sufficient to induce synaptic phenotypes and cytotoxicity even in Healthy motoneurons, indicating that loss of CREB activity may be the cause, rather than an epiphenomenon, of MN vulnerability. Thirdly, we now show that the CREB pathway is not only affected in terms of CREB phosphorylation (well before MN death), but also in terms of sequestration of the CREB co-factor CBP within aggregates.

Since similar pCREB^{S133} dynamics are detected in TBK1-mutant MN, we hypothesized that the presence of aberrant aggregates, irrespective of the mutant gene involved, might be involved in CREB dysregulation. Indeed, not only loss of CREB phosphorylation is shared by C9orf72- and TBK1-mutant MN, but the aggregation of CBP is observed in ALS^{TBK1} MN, too (shown in Figure 1I and Figure EV3A).

Major points:

1. In Fig1A, what was the stage when the authors performed RNAseq in Fig1A? Did they perform the experiments before the onset of MN degeneration or after MN degeneration? This hugely influenced the conclusion if the dysregulated genes in Fig1B~C due to disease manifestation or merely a consequence of dying of MNs.

>> We have now clarified in the text that the RNAseq shown in Fig 1 was performed at DIV35 (line 136), when no MN loss in the ALS cultures was detected. Thus, these transcriptomes represent not an effect of cellular suffering but rather disturbances that pre-date the cell death.

2. In many cases, the authors showed the data without controls (Fig 1H, FigEV1B, EV1E, EV1F), or no representative figures were shown to back up the statistical quantifications. It is difficult for me to evaluate the data and conclusion.

>> We show now the quantification of the corresponding controls and representative pictures for each experiment within the figures. It must be stressed that controls were taken into account during the statistical analysis and were not shown only to simplify the layout of the figures.

3. The application of optogenetics in Fig2 is great to elicit neuron activation, yet the readout with poly(GA) aggregation is insufficient. The quantification of MN numbers or survival upon this optogenetic manipulation, or the level of pCREB, will be more direct access to the phenotype after this manipulation.

>> In the revised manuscript we now show that optogenetic manipulation increases the levels of pCREB^{S133} in hiPSC-derived MN (as well as in poly(GA)-expressing primary neurons). We found that the levels of pCREB^{S133} were increased after 1 and 10 Hz stimulation protocols, in direct correlation with the neuroprotective effect previously observed. This data is presented in Figure EV4B (hiPSC-derived MN) and Appendix Figure S6B and S6D (primary neurons).

4. The statistical method and the figure legend shall be enhanced to provide readers with correct information. For example, it not clear to me how was the survival index in FigEV1B is calculated? This could be applied to many of the supplementary figures.

>> The figure legends have been improved following the guidelines of EMBO Molecular Medicine. Since similar doubts regarding our survival index have been also raised by Reviewer #2, we now provide a direct assessment of MN survival, measured in terms of number of ChAT+ cells per area. We further characterized their morphology using a standardized quantification of neurite length. Both read-outs provide converging evidence of the beneficial effect of Apamin and XE991 on MN survival and on their structural integrity.

Minor points:

The authors might want to revise several sentences, including lines 77~80, 83,92, 141~148, 174~176, 193~197 as I found it quite challenging to digest.

>>The manuscript has been streamlined to improve readability.

Referee #2 (Remarks for Author):

Catanese et al., investigated transcriptional profiles of human iPSC-derived motor neurons from 3 patients with mutations in C9orf72, a patient with TBK1 mutation, a patient with TBK1/FUS mutations, a mutation-corrected control of C9orf72, and two healthy controls. They found that CREB target genes were commonly altered in motor neurons from ALS patients compared to controls which was accompanied with downregulation of synapse and ion-channel associated genes. The expression levels of pCREB in motor neurons were elevated at DIV21 but decreased at DIV 70. These findings were recapitulated in rat primary cortical neurons in which EGFP-poly (GA) was overexpressed, which exhibited similar up- and down-regulation of pCREB as well as a decrease in pre-synapse numbers and morphological alteration of post-synapse. The authors also found that 1 Hz stimulation which caused channel rhodopsin-dependent activation of motor neurons decreased the amount of poly-GA but 10 Hz stimulation increased, suggesting neuronal firing was involved in C9-related ALS pathogenesis. Finally, they investigated whether selective K⁺ channel blockers to ameliorate pathological readouts. They found that both Apamin and XE-991 rescued decreased pCREB expression, altered Creb-target gene expressions, and mitigated synaptic deficits in C9-motor neurons.

Overall, the study is well-written and the findings that K⁺-channel mediated neuronal firing is associated with vulnerability is of intrigue. However, the manuscript is still immature, and there are many issues to be addressed before publication as listed below.

Major issues

1. The data from rat primary cortical neurons were confusingly mixed with those from human iPSC-derived motor neurons. It is necessary to reconsider the order of figures. For instance, if all the data from rat cortical neurons are moved to supplementary figures, the manuscript would be more straight forward.

>> In line with the Reviewer's suggestion (and of Reviewer #3 as well), we moved the data obtained with primary neurons to supplementary figures (Appendix). To further streamline the manuscript, we also removed the data regarding the TBK1-FUS line, and moved the results from TBK1-mutant MN to separated Expanded View Figures. Although the main figures have been streamlined to concentrate the focus on C9 neurons, we believe that the data on TBK1-mutant MN reveal important convergent pathologic features shared by distinct mutations associated with familial cases of ALS/FTD. Therefore, we have maintained the text describing the TBK1 data and the relative discussion.

2. In Fig1, the authors found that 1001 from 1766 differentially expressed genes were Creb-targeted. They concluded that it was significantly higher, but they did not show how much population of Creb-targeted genes were in general.

>> We apologize for the confusing presentation of the data. The statistical significance obtained with the PASTAA algorithm does not refer to a specific number or population of genes, rather to the association of a specific transcription factor with the expression of the gene set of interest.

In the revised manuscript we have now clarified that the association of this transcription factor to the transcriptome of C9orf72-mutant MN is statistically significant. We now report the p-value obtained with the PASTAA algorithm, matching the data displayed in Figure 3C.

3. All graphs must be shown with individual blots of control (vehicle) instead of dashed lines.

>> The graphs have been modified accordingly.

4. The mechanism is not clearly explained how mutations in C9orf72 and subsequent poly-GA accumulation and TBK1/FUS accumulation lead to upregulation of pCreb in nuclei followed by alteration of Creb-regulated genes.

>> The mechanisms connecting mutations in C9orf72 and TBK1 first to the up-regulation (and, likely, of intrinsic hyperexcitability) and later on to the down-regulation of pCREB are the subject of active investigation. Several hypotheses, not mutually exclusive, are currently taken into

consideration. Since both mutations lead to the formation of cytoplasmic aggresomes, and a similar phenotype is triggered by poly(GA) overexpression, we believe that the aggresomes themselves (or the associated disruption in autophagy, as shown by Catanese et al., 2019) might alter CREB activity. In the revised version of the manuscript we report one additional mechanism which may link aggregate formation to the altered expression of synaptic genes: the aggresomes detected in ALS^{C9orf72} and ALS^{TBK1} MN sequester the CREB coactivator CBP, which is required for an efficient transcriptional activity of CREB. Thus, loss of co-regulators contributes to the overall decrease in CREB-regulated genes, along with the decrease in pCREB itself (please see the answer to Reviewer 1 as well).

5. Did the authors observe poly-GA accumulation in hiPSC-derived MNs with C9 mutations? If so, they can treat those motor neurons with Apamin and XE-991 to see their effects on poly-GA accumulation. If not, they need to explain the rationale to use poly-GA overexpression system in rat cortical neurons.

>> We mention now within the text that we used poly(GA) because this is the most abundant product produced by RAN translation in presence of a C9orf72 mutation (lines 206-207). Regarding our human MN, we did not detect any GA aggregate by immunofluorescence at the time points investigated. It must be stressed that, to date, such detection has been reported and quantified only by one group (Pal et al., 2021) at a maturation stage more advanced than the one used here. Thus, detection of GA aggregates in human MN is not yet considered uncontroversial. We have nevertheless expanded the scope of our investigation of the effects of neuronal excitation on the pathogenic pathways set in motion by the pathogenic expansion in the C9orf72 gene. We now show in Figure 2F that treatment with Apamin and XE991 reduced the occurrence of RNA foci, i.e. reduced the accumulation of the RNA encoded by the hexanucleotide repeats.

6. Since the authors successfully identified aggresome formation in iPSC-derived motor neurons from a case of ALS-TBK1, they can see whether Apamin and XE-991 have beneficial effects in TBK1-MNs. It would validate their conclusion that K⁺-channel mediated neuronal firing is involved in ALS pathogenesis in general.

>> We have performed the experiment suggested by the Reviewer, and the results are shown in Figure EV 5C. In line with the reduction of SQSTM1 aggregates in C9orf72 and TBK1 MN upon treatment, exposure to Apamin and XE991 reduced the size of aggresomes also in ALS-TBK1 MN.

7. In experiments of poly-GA in rat cortical neurons, the authors used AAV to overexpress EGFP-poly-GA in primary neurons. In general, AAVs are good infection efficiency to express genes or shRNAs in vivo, but not as good as lentivirus for in vitro. It is necessary to show the expression ratio of EGFP-poly-GA positive neurons.

>> We have quantified the infection efficiency of our AAV9 in cell culture. In line with previously published data (Royo et al., 2008), almost 80% of the neurons in primary cultures were infected. We have added this information in Appendix Figure S4A.

8. In Fig2A-C, it would be more convincing if the authors chase the EGFP-poly GA-positive aggresomes of primary neurons in living before/after 1 or 10Hz stimuli.

>> We agree with the reviewer that this would be an interesting approach to pursue. Unfortunately, we do not have access to a live cell imaging setup that might be used in combination with optogenetic devices to perform such an interesting experiment.

9. To eliminate the possibility that drugs and/or optogenetic stimuli can affect transcription of EGFP-poly-GA, mRNA levels should be also measured.

>> We are thankful to the Reviewer for raising this point. To our surprise, we found that Apamin and XE991 drastically reduced the expression of EGFP-poly(GA). Since the drugs also reduced the expression of EGFP to a similar extent (in cultures expressing the EGFP control AAV9), we wondered which mechanism might lead to the reduced expression of these constructs. We speculated that the increased activity induced by the K⁺ channel blockers might induce RNA degradation. In fact, neuronal firing alters the levels of mRNA depending also on the GC content (Kiltschewskij et al., 2020). We further explored if the increased neuronal firing may affect the stability of the endogenous

mutant C9 RNA in the iPSC MN. We performed FISH against GGGGCC repeats and analysed the accumulation of RNA foci in ALS^{C9orf72} MN upon treatment with Apamin and XE991. We found that both drugs significantly reduced the number of foci in mutant MN (Fig 2F), suggesting that enhancing MN firing also beneficially affects the levels of RNA foci. Indeed, neuronal firing regulates the accumulation of RNA granules (Wong et al., 2021). We believe that these new data further strengthen the neuroprotective effect of Apamin and XE991.

10. It is necessary to confirm the neuronal activities by other methods like c-fos expression.

>> We have monitored the neuronal activity using c-Fos, as recommended by the reviewer. We performed immunostaining against c-Fos in C9orf72-mutant human MN (shown in Appendix Figure S10): while we could barely detect any c-Fos signal in vehicle-treated cultures, almost all of the MN exposed to Apamin and XE991 showed a clear and intense nuclear signal, indicating a successful increase in neuronal firing.

11. It is necessary to explain why the authors used stimulation frequencies, 1 and 10Hz for experiments.

>> We clarified now, within the Methods section, that these frequencies were chosen based on existent literature analysing the electrophysiological properties of neurons in different models of ALS and SMA, including hiPSC-derived MN. Indeed, Wainger and colleagues have shown that MN from healthy controls fire at a frequency of 1.17 Hz (Wainger et al., 2014).

12. In the experiment of Fig.2C, was archaerhodopsin change the amount of poly-GA accumulation?

>> We show now that archaerhodopsin overexpression per se, without activation, did not alter the accumulation of poly(GA) when compared to a control construct expressing Td-tomato, and did not cause neuronal loss as well (Appendix Fig S8). Moreover, we show that the aggregate load detected in ArchT-positive apoptotic neurons upon light stimulation is larger than in surviving neurons (Appendix Fig S7F).

13. The authors should describe the motor neuron purity ratio by immunostaining using specific antibodies for neurons and glial cells. For instance, the signals of cleaved PARP might be in glial cells In Fig EV4B.

>> We now include the characterization of the MN cultures in new Appendix Figure S1, showing that glial cells cover a very small portion of the cultures.

Since it has been recently shown that PARP expression is low in spinal motoneurons, but increased in astrocytes of ALS patients (Kim et al, 2003), we have further confirmed the findings regarding the apoptotic cell death using cleaved caspase-3 as second, independent marker. We analyzed the number of cleaved caspase 3+/CHAT+ cells in vehicle- and K⁺ channel blockers-treated cultures (shown in Appendix Figure S11). We confirmed here that Apamin and XE991 reduced the number of cleaved caspase-3+ MN, in agreement with the PARP data, confirming the neuroprotective effect of Apamin and XE991.

14. The "survival index" used in this study is not convincing, since it represents only the ratio of MAP2 or CHAT-positive area in each image, which can be affected by morphological changes or even immunostaining or culture conditions. There are more reliable alternative indices such as neurite outlength or neurite blanch number which can be automatically measured by software.

>> According to the Reviewers' suggestions (see also the points raised by Referee #1), we provide now a direct assessment of MN survival over time in terms of number of CHAT+ neurons in cultures (as shown in Fig 1G), and, independently, we evaluated the neuroprotective effect of the Apamin and XE991 on the structural integrity of MN by analyzing the neurite length using the semi-automated SNT plugin of the Neuroanatomy package for ImageJ (as shown in Fig 2A). Both analyses confirm a substantial increase in MN survival and enlarged dendritic arborization upon Apamin and XE991 treatment.

Minor issues

1. From line 157 to 179, Fig 3B-3J must be Fig 2B-2J.

>> *We have revised the whole manuscript and corrected the figure calls.*

2. The data of Fig 2E-J which show the effects of candidate drugs should be separated from Fig 2A-D.

>> *The original figure has been split in 2 new figures: data on primary neurons is shown now in Appendix Figure S6, and the optogenetic data in human MN moved to the Figure EV4.*

3. In Fig 1B It would be helpful if Creb-target genes are specified by different color.

>> *We have revised the format of Fig1 to increased readability in the context of the new data added for the revision. As such, we have replaced the volcano plot with a table showing the statistical association of the PASTAA-identified transcription factors (including CREB) to the altered transcriptome of C9 MN.*

4. In Fig 1E immunostaining is necessary to be shown. Similarly, it should be shown in Fig 1J as well.

>> *We have added representative pictures of the immunostainings.*

5. The description of *** $p < 0.001$ is missing in figure legends.

>> *Thanks for spotting this, the figure legends have been improved and corrected.*

6. In L770 "Scale bars: E-F 25um" must be "Scale bars: F 25um".

>> *We have corrected the mistake, and modified the format of the figure legends according to the author guidelines of the journal.*

7. The staining images of Apamin should be included in Fig. 2G and 2I.

>> *Images have been added.*

8. Scale bar is needed in Fig. 3F.

>> *The scale bar has been added.*

9. Arrows in Fig. EV1B and EV1F should be explained in legends.

>> *The figure legends have been improved accordingly.*

10. The description of *** $p < 0.001$ should be omitted in the legend of Fig. EV3.

>> *We have omitted the description.*

Referee #3 (Comments on Novelty/Model System for Author):

This reviewer does have a major concern regarding the initial approach to transcriptomics presented as Figure 1, as is indicated below. The inclusion of multiple ALS mutant lines in some but not all Figures is confusing, particularly since the authors have chosen to identify all as ALS-MN. For the results and conclusions drawn to have more scientific merit, it is the recommendation of this reviewer to only include C9 patient lines, indicate specifically which line produces each result within figures, and increase the number of lines utilized to have appropriate statistical power.

Referee #3 (Remarks for Author):

In this article, the authors evaluate the transcriptomics of several ALS-patient derived motor neuron lines. Initially, they identify synaptic protein dysregulation, with pCREB levels over time being a major driving factor of these differences. They screen several compounds known to modulate neuronal excitability, and identify Apamin and XE991 as capable of reducing poly-GA DPR burden in transduced primary cortical neurons. Using these compounds in ALS-MN, they then showed transcriptomic and protein profiles, along with upregulation of pCREB and synaptic components Basoon and Homer compared with vehicle conditions.

Overall, the authors propose that this work provides a framework for modulation of potassium channels in mitigating DPR burden in C9-ALS patient cells, by restoring synaptic activity and restoring autophagic processing.

This reviewer does have a major concern regarding the initial approach to transcriptomics presented as Figure 1, as is indicated below. The inclusion of multiple ALS mutant lines in some but not all Figures is confusing, particularly since the authors have chosen to identify all as ALS-MN. For the results and conclusions drawn to have more scientific merit, it is the recommendation of this reviewer to only include C9 patient lines, indicate specifically which line produces each result within figures, and increase the number of lines utilized to have appropriate statistical power.

The remaining concerns of this reviewer revolve around lack of details for clarity of interpretation.

Strengths:

The supporting evidence for autophagy impairment presented as Figure EV1 contains very striking images and robust phenotypic outcomes.

The presentation of GO analysis terms in Figure 1 C and D are well designed and organized. They allow the reader to easily come to the conclusions made by the authors in-text.

The data presented in Figure EV4 is very promising that the compounds Apamin and XE991 may be effective in increasing survival and reducing autophagic burden in C9 patient-derived neurons. Particularly, it is informative it is the functionality, rather than the expression levels of these channels that are altered.

The Materials and Methods section is extremely detailed and thorough.

We are grateful to the reviewer for this positive feedback.

Major points of concern:

A major concern for this paper lies in the initial transcriptomics evidence provided in Figures 1A and 1B. There are dramatically significant differences between the two C9 patient-derived lines presented. Furthermore the "correction" of the phenotype from line ALS-C9orf72II to Corrected still has profound differences when compared with the healthy line shown. Lumping the results from only these 2 conditions in each case, and determining that is typical of C9 or "Healthy" seems to be an over representation of transcripts that are altered due to very limited biological sampling as per a power analysis justification. It is recommended that either C9 lines are compared with their own isogenic partners, or an increased line number of at least 5 per condition is required to make such conclusions.

>> To address this point, we re-arranged the design of the RNA-seq analysis to circumvent the differences between cell lines. This approach, based on the direct comparison of the ALS and Healthy groups, has two main advantages: 1) it addressed the sample size limitation concern raised by the reviewer (each group has now 6 samples) 2) it robustly reflects the main transcriptional differences between the genotypes, independent of the confounding effect coming from different cell lines. Importantly, this approach confirmed the core results highlighted by our previous analysis.

Considering that the Reviewer expressed some concerns regarding our data as being not fully representative of typical C9orf72 cases, we performed Gene Set Enrichment Analysis (GSEA). We focused this supervised analysis on synaptic alterations, which are the findings of main interest for this manuscript. By this approach, we compared our RNAseq data to previously published transcriptomes, with the aim of gaining deeper insights into the altered expression of synaptic genes in C9 MN and increasing the translational relevance of our results. This new supervised analysis, shown in Appendix Figure 2, revealed an up-regulation of pathways involved in neuronal stress and aging, and confirmed the downregulation of synaptic activity-related pathways already highlighted in the unsupervised analysis shown in main Fig 1. Both the unsupervised and supervised analysis, with their strong convergence, support the view of a dramatically altered synapse-related transcriptome in ALS-C9orf72 MN (also in agreement with Sareen et al, 2013; Selvaraj et al, 2018). Furthermore, an independent study (Mehta et al. 2021) compared the transcriptomes of C9 hiPSC-derived motoneurons to those of their respective isogenic controls, and found similar down-regulation in the expression of genes involved in synaptic transmission and plasticity. Thus, the congruence of our data with independent studies performed by using unrelated hiPSCs indicates altered expression of synaptic transcripts as a pathological manifestation of C9orf72-mutant MN. Furthermore, we confirmed the reduced expression of synaptic genes using targeted single-tube qPCR experiments using all the C9 and Healthy lines: the expression of NLGN3, SLC6A, TRIM9, SYT1, SYNGR1, and SNAP91 was in fact significantly lower in C9 than control MN (new Figure 1F).

It must be stressed that comparison of mutant lines with their own isogenic CRISPR-corrected controls is fraught with biases. In fact, correction of C9orf72 mutation does not resemble the transcriptional landscape of healthy individuals, as revealed by our own data (and noticed by the reviewer) and confirmed by the strong segregation of the transcriptomes of C9 neurons and their corresponding isogenic controls reported by Perkins et al. (2021). These observations lean in favour of our approach of including in the initial RNAseq MN from a healthy donor not related to our mutant lines as well.

Although we did not have access to further C9 lines as suggested by the Reviewer, the new data produced during the revision period confirmed and strengthened the results revealing a significantly reduced expression of synaptic genes in ALS^{C9orf72} MN, and the reliability of our initial RNAseq.

Overall it is not clear why the TBK1 and TBK1/FUS lines were included, as the paper is focusing on the C9 phenotype, the transcriptomics and proteomics are only done with C9 comparisons, and the TBK1/FUS lines were not presented in all figures. If additional lines were needed in order to attain statistical comparison, additional C9 lines should have been utilized. Doing so will also clear up some confusion throughout the entire manuscript, as the authors refer to all lines as ALS MN, but the phenotypes and cellular pathways involved in C9, TBK1, and FUS mutations are distinct. Presently it is difficult to draw conclusions when the exact genotype is not known. Furthermore, throughout all figures it would be beneficial to somehow distinguish between the different C9 lines. This would also

allow for a comparison of C9 repeat expansion lengths for the results shown. Additionally, as in Figure 3E, it would allow the reader to see if the lines respond differentially to the different compounds.

>> *We streamlined and reorganized the structure of the manuscript to highlight the data on the C9 mutant lines, separately labelling them within the graphs with different symbols. The data relative to the mutant TBK1 MNs have been moved to separate Expanded View figures (where we compared TBK1 cells to an age-matched control, or to vehicle in case of treatments with Apamin and XE991). Nevertheless, the comparison between C9 and TBK1 MN remains, in our view, valuable in underscoring the converging pathomechanisms triggered by mutations in different genes and the equally beneficial effect of the K⁺ channel blockers.*

The shift in timecourse from the initial transcriptomics assessment in Figure 1 (28 DIV) to that performed in Figures 3 (DIV70) was not described or rationalized. Additionally, the specific lines that were treated and compared are not indicated, presumably comparisons were performed for the same line in vehicle and treatment conditions.

>> *We clarify now that the initial RNAseq was performed at DIV 35, a time point where a clear phenotype is detectable in the ALS lines (aggregates accumulation), but still preceding neuronal loss. The second RNAseq was performed at DIV 70 because at this stage of culture we detected a clear loss of MN and synapses in all the C9orf72 lines, and these phenotypes are rescued by Apamin and XE991. Thus, we applied the same treatment design by exposing the C9orf72 cultures for 7 days to Apamin or XE991 (starting from DIV63) before performing RNAseq in order to analysing the neuroprotective transcriptional program activated by K⁺ channel blockade.*

Also, we now indicate in the figure legends which hiPSC lines were used in the two RNAseq: at DIV35 we used ALS^{C9orf72}I and ALS^{C9orf72}II, while at DIV70 ALS^{C9orf72} II and ALS^{C9orf72} III.

Minor points of concern:

The introduction was thorough, it would be further strengthened by inclusion of a sentence explicitly stating how manipulating K⁺ channels would alter neuronal excitability.

>> *Thanks for this suggestion, the required information has been added (lines 103-106).*

It is unclear why the authors chose to use primary cortical neurons rather than primary motor neurons (or healthy IPS) for their transduction/transfection experiments.

>> *We agree with the reviewer, using MN from healthy donors might provide data more translationally relevant. Unfortunately, although we originally tried to overexpress poly(GA) in healthy iPSC-derived MN, our AAV vector did not work in this model. For this reason, we decided to use primary cortical cultures, since this model was used in the original publication of May and colleagues (2015). We found this model also appropriate for our scope since several patients with C9orf72 and TBK1 mutations are affected by FTD. Nevertheless, since the main focus of the paper is the effect of Apamin and XE991 on human MN, we moved the data obtained with primary cultures to supplementary figures.*

· The formation of poly(GA)-SQSTM1/p62 cytoplasmic inclusions (Fig 1H) and reduction in number of primary dendrites (Fig1I) has already been shown by May et al 2014.

>> *We intentionally repeated the experiments already published by May and colleagues (2015) in order to provide an internal quality control for our newly-generated AAV9-hSyn-poly(GA) construct. We did not intend to present these as innovative findings (they do not constitute the core of our findings) and we have now clearly stated it within the manuscript.*

· The authors state on line 162 "to further assess whether restoration of MN activity", however the experiments they then describe are in primary cortical neurons.

>> *We have corrected this mistake.*

Figure 1J/K would be strengthened by addition of DIV14 images contrasted with DIV28 images, to show reduction of pCREB visually to the audience.

>> *The images at DIV14 have been added as requested.*

The transition from line 121 "next assessed if ALS MN would also display synaptic abnormalities" to line 122 "indeed pharmacological blockade" in healthy MN does not match up.

>> *We have rephrased the sentence.*

For Figure EV3D- were all 3 C9 and healthy lines assessed at each timepoint? It is unclear from the figure legend and presentation of the Western blot included. This is a point of concern given the spread of phenotype shown in figure EV3A. If for instance the top healthy line and TBK1 mutant were compared the effect size would be much greater than the corrected C9 and highest point C9-ALS.

>> *All the 3 C9 and healthy lines were used for the experiment at each timepoint. We have now modified the graph showing each line with separated symbols.*

The loss of synaptic components presented in Figure EV3 E-K is very interesting. However, it seems that ~20 cells/points were sampled in E and ~30 sampled for I. An equal number should be evaluated for both, particularly since there seems to be a trend at DIV14 that may be significant if additional values were assessed. This applies to all figures, as the number of cells evaluated in Figure 2A-D is also not indicated.

>> *We corrected this mistake and now show the graph with an equal number of neurons analysed. The results remained similar to those shown in the previous version.*

The results in Figure 2D are indicated as "hiPSC-derived MN" both in-text, and in-figure. Within the figure legend it says "ALS MN", which the reviewer is assuming is one of the C9 lines. It would be beneficial to explicitly state this in the text and on the figure.

>> *The figure legend has been amended accordingly.*

It is unclear why Figure 3F does not include all 3 C9 lines. Furthermore, an evaluation of Apamin and XE991 treatments in healthy MNs is lacking. It will be meaningful to determine if the upregulations seen approach "normal" levels of synaptic components.

>> *We have repeated the experiment including all the C9 and Healthy lines. We now show that treatment with Apamin and XE991 elevates the density of synapses in C9orf72-mutant MN (although still lower than the one in Healthy lines), but does not have any effect on the synapses of Healthy cells.*

As there are ~250 proteins differentially altered between Apamin and XE991 treatments, would a combination be something to consider as well?

>> *Despite this being an intriguing point, we believe that mixing the two molecules might require extensive testing to find a balanced concentration of both drugs not inducing neuronal loss. In fact, Apamin and XE991 target two different subtypes of K⁺ channels, which control two different currents. We believe that the combinatory effect of blocking Kv7 and SK channels at the same time might increase MN firing to an extent resulting in detrimental effects for the neurons. In fact, our data suggest that the neuroprotective effect of increased activity depends on the dose/concentration of the molecules used: high concentration of XE991 induced apoptosis in poly(GA)⁺ neurons, and 10Hz increases the accumulation of aggregates. For these reasons, although the approach suggested by the reviewer might be theoretically feasible, we believe it goes beyond the aim of this paper.*

Overall, the paper would benefit from a proofreading to rectify the following small errors and potentially more that the authors will identify:

· It is standard practice for graphs to be plotted mean {plus minus} SEM with error bars in both directions. It is not indicated whether this is the case. Additionally, it would be beneficial for the reader to present these error bars with hatch marks instead of just lines.

>> *We have modified the data display following this suggestion, and state within the figure legends that the error bars represent SEM.*

· The "purple triangle healthy" indicated in the top of Figure 1A is not represented on the figure legend at the right hand side of that panel.

>> *We apologize for this inconvenience, which we believe originated when the figure was uploaded. To avoid this and similar issues, we show now a black-purple bar instead of the triangle, whose larger size should prevent any alteration during file transfer.*

· The circles in Figure EV3B-C are too small to distinguish their colors & therefore genotypes.

>> *We have increased the size of the symbols.*

· Figure 2B-H are mislabeled as C in the text.

>> *We have deeply reorganized the paper and the sequence of the Figures, according to the Reviewers' suggestions listed above.*

· Figure 2F is missing indicators of statistical significance for compounds that were efficacious.

>> *The information has been added.*

References

- Devlin AC, Burr K, Borooh S, Foster JD, Cleary EM, Geti I, Vallier L, Shaw CE, Chandran S, Miles GB (2015) Human iPSC-derived motoneurons harbouring TARDBP or C9ORF72 ALS mutations are dysfunctional despite maintaining viability. *Nat Commun.* 6:5999. doi: 10.1038/ncomms6999.
- Kiltschewskij DJ, Cairns MJ (2020) Transcriptome-Wide Analysis of Interplay between mRNA Stability, Translation and Small RNAs in Response to Neuronal Membrane Depolarization. *Int. J. Mol. Sci.* 21(19):7086. doi: 10.3390/ijms21197086.
- Kim SH, Henkel JS, Beers DR, Sengun IS, Simpson EP, Goodman JC, Engelhardt JI, Siklós L, Appel SH (2003) PARP expression is increased in astrocytes but decreased in motor neurons in the spinal cord of sporadic ALS patients. *J Neuropathol Exp Neurol.* 62(1):88-103. doi: 10.1093/jnen/62.1.88.
- Leroy F, Lamotte d'Incamps B, Imhoff-Manuel RD, Zytnicki D (2014) Early intrinsic hyperexcitability does not contribute to motoneuron degeneration in amyotrophic lateral sclerosis. *Elife.* 3: e04046. doi: 10.7554/eLife.04046.
- Martínez-Silva ML, Imhoff-Manuel RD, Sharma A, Heckman CJ, Shneider NA, Roselli F, Zytnicki D, Manuel M (2018) Hypoexcitability precedes denervation in the large fast-contracting motor units in two unrelated mouse models of ALS. *Elife.* 7: e30955. doi: 10.7554/eLife.30955.
- May S, Hornburg D, Schludi MH, Arzberger T, Rentzsch K, Schwenk BM, Grässer FA, Mori K, Kremmer E, Banzhaf-Strathmann J, *et al* (2014) C9orf72 FTLN/ALS-associated Gly-Ala dipeptide repeat proteins cause neuronal toxicity and Unc119 sequestration. *Acta Neuropathol.* 128(4): 485–503. doi:10.1007/s00401-014-1329-4.
- Mehta AR, Gregory JM, Dando O, Carter RN, Burr K, Nanda J, Story D, McDade K, Smith C, Morton NM, *et al.* (2021) Mitochondrial bioenergetic deficits in C9orf72 amyotrophic lateral sclerosis motor neurons cause dysfunctional axonal homeostasis. *Acta Neuropathol.* 141(2):257-279. doi: 10.1007/s00401-020-02252-5.
- Naujock M, Stanslowsky N, Bufler S, Naumann M, Reinhardt P, Sternecker J, Kefalakes E, Kassebaum C, Bursch F, Lojewski X, *et al.* (2016) 4-Aminopyridine Induced Activity Rescues Hypoexcitable Motor Neurons from Amyotrophic Lateral Sclerosis Patient-Derived Induced Pluripotent Stem Cells. *Stem Cells.* 34(6):1563-75. doi: 10.1002/stem.2354.
- Wainger BJ, Kiskinis E, Mellin C, Wiskow O, Han SS, Sandoe J, Perez NP, Williams LA, Lee S, Boulting G, *et al.* (2014) Intrinsic membrane hyperexcitability of amyotrophic lateral sclerosis patient-derived motor neurons. *Cell Rep.* 7(1):1-11. doi: 10.1016/j.celrep.2014.03.019.

- Pal A, Kretner B, Abo-Rady M, Glaß H, Dash BP, Naumann M, Japtok J, Kreiter N, Dhingra A, Heutink P, *et al.* (2021) Concomitant gain and loss of function pathomechanisms in C9ORF72 amyotrophic lateral sclerosis. *Life Sci Alliance*. 4(4):e202000764. doi: 10.26508/lsa.202000764.
- Perkins EM, Burr K, Banerjee P, Mehta AR, Dando O, Selvaraj BT, Suminaite D, Nanda J, Henstridge CM, Gillingwater TH, *et al.* (2021) Altered network properties in C9ORF72 repeat expansion cortical neurons are due to synaptic dysfunction. *Mol Neurodegener*. 16(1):13. doi: 10.1186/s13024-021-00433-8
- Royo NC, Vandenberghe LH, Ma JY, Hauspurg A, Yu L, Maronski M, Johnston J, Dichter MA, Wilson JM, Watson DJ (2008) Specific AAV serotypes stably transduce primary hippocampal and cortical cultures with high efficiency and low toxicity. *Brain Res*. 1190:15-22. doi: 10.1016/j.brainres.2007.11.015.
- Wong CE, Jin LW, Chu YP, Wei WY, Ho PC, Tsai KJ (2021) TDP-43 proteinopathy impairs mRNP granule mediated postsynaptic translation and mRNA metabolism. *Theranostics* 11(1): 330-345. doi:10.7150/thno.51004.

10th May 2021

Dear Dr. Catanese,

Thank you for the submission of your revised manuscript to EMBO Molecular Medicine. I am pleased to inform you that we will be able to accept your manuscript pending the following final amendments:

- 1) Please address all referees' points.
- 2) In the main manuscript file, please do the following:
 - Correct/answer the track changes suggested by our data editors by working from the attached/uploaded document.
 - Add up to 5 keywords.
 - Make sure that all special characters display well.
 - Remove font color.
 - Remove "data not shown" (p. 9).
 - Remove the abbreviations list and incorporate abbreviations into the text.
 - In M&M, provide the antibody dilutions that were used for each antibody.
 - In M&M, provide a statistical paragraph that should reflect all information that you have filled in the Authors Checklist, especially regarding randomization, blinding, replication.
 - Include a statement that informed consent was obtained from all human subjects and that, in addition to the WMA Declaration of Helsinki, the experiments conformed to the principles set out in the Department of Health and Human Services Belmont Report.
 - Indicate in legends exact $n=$ and exact $p=$ values, not a range, along with the statistical test used. To keep the figures "clear" some authors found providing an Appendix table Sx with all exact p -values preferable. You are welcome to do this if you want to.
 - In addition to the accession number please provide URL for deposited datasets. Please be aware that all datasets should be made freely available upon acceptance, without restriction. Use the following format to report the accession number of your data:

The datasets produced in this study are available in the following databases:
[data type]: [full name of the resource] [accession number/identifier] ([doi or URL or identifiers.org/DATABASE:ACCESSION])

Please check "Author Guidelines" for more information.

<https://www.embopress.org/page/journal/17574684/authorguide#availabilityofpublishedmaterial>

3) Funding: Please make sure that information about all sources of funding are complete in both our submission system and in the manuscript.

4) Synopsis: Every published paper now includes a 'Synopsis' to further enhance discoverability. Synopses are displayed on the journal webpage and are freely accessible to all readers. They include separate synopsis image and synopsis text.

- Synopsis image: Please provide a striking image or visual abstract as a high-resolution jpeg file 550 px-wide x (250-400)-px high to illustrate your article.

- Synopsis text: Please provide a short stand first (maximum of 300 characters, including space) as well as 2-5 one sentence bullet points that summarise the paper as a .doc file. Please write the bullet points to summarise the key NEW findings. They should be designed to be complementary to the abstract - i.e. not repeat the same text. We encourage inclusion of key acronyms and quantitative information (maximum of 30 words / bullet point). Please use the passive voice.

5) For more information: There is space at the end of each article to list relevant web links for further consultation by our readers. Could you identify some relevant ones and provide such information as well? Some examples are patient associations, relevant databases, OMIM/proteins/genes links, author's websites, etc...

6) As part of the EMBO Publications transparent editorial process initiative (see our Editorial at <http://embomolmed.embopress.org/content/2/9/329>), EMBO Molecular Medicine will publish online a Review Process File (RPF) to accompany accepted manuscripts. This file will be published in conjunction with your paper and will include the anonymous referee reports, your point-by-point response and all pertinent correspondence relating to the manuscript. Let us know whether you agree with the publication of the RPF and as here, if you want to remove or not any figures from it prior to publication. Please note that the Authors checklist will be published at the end of the RPF.

7) Please provide a point-by-point letter INCLUDING my comments as well as the reviewer's reports and your detailed responses (as Word file).

I look forward to reading a new revised version of your manuscript as soon as possible.

Yours sincerely,

Zeljko Durdevic

***** Reviewer's comments *****

Referee #1 (Remarks for Author):

The authors have addressed all of my concerns, and have performed a substantial amount of additional experiments to address all reviewers' concerns. I, therefore, endorse the publication of this manuscript.

Referee #2 (Remarks for Author):

The manuscript is quite improved and now suitable for publication by addressing many issues raised in the 1st review. I just have a minor comment.

It would be helpful for readers if the authors mention that it is difficult to detect GA aggregates in human motor neurons regarding to the comment #5.

Referee #3 (Comments on Novelty/Model System for Author):

In this round of revisions, the authors have satisfactorily addressed the concerns and comments of this reviewer.

Referee #3 (Remarks for Author):

Few minor corrections/revisions are detailed below.

line 121-123 Olig2 positive cells could be oligodendrocyte precursors in addition to motor neuron precursors. Additionally, the CC1 antibody is typically used to label oligodendrocytes in tissue; in vitro the marker O4 would be preferred for all cells of the oligodendrocyte lineage, while A2B5 would be specific oligodendrocyte precursors and MBP or PLP for more mature myelinating cells.

lines 128-130 TEM morphological similarities of C9 to TBK1 mutant are informative, and the cells do look similar to each other, but it but difficult to assess the importance without a "normal" control as baseline for comparison.

some of the error bars in Fig 1G quantification seem to be right-shifted, as they present align between the c9 of one timepoint and the controls of the next time.

Line 194- A rationale is required as to why TBK1 was chosen here. Since it replicates all of the effects, readers may question if this would be true of any/all ALS lines. Either another line that has negative results is needed, or a strong rationale is needed as to why TBK1 specifically would be anticipated to have a similar phenotype. Otherwise, it is difficult to determine if this finding is applicable to ALS as a whole, or potentially a subset of genetic categories.

***** Reviewer's comments *****

Referee #1 (Remarks for Author):

The authors have addressed all of my concerns, and have performed a substantial amount of additional experiments to address all reviewers' concerns. I, therefore, endorse the publication of this manuscript.

>> We want to thank the Reviewer for the positive feedback and constructive comments.

Referee #2 (Remarks for Author):

The manuscript is quite improved and now suitable for publication by addressing many issues raised in the 1st review. I just have a minor comment.

>> We want to thank the Reviewer for the positive feedback and constructive comments.

It would be helpful for readers if the authors mention that it is difficult to detect GA aggregates in human motor neurons regarding to the comment #5.

>> We mention now within the manuscript that the detection of GA aggregates in iPSC-derived neurons has been inconsistent in literature.

Referee #3 (Comments on Novelty/Model System for Author):

In this round of revisions, the authors have satisfactorily addressed the concerns and comments of this reviewer.

>> We want to thank the Reviewer for the positive feedback and constructive comments.

Referee #3 (Remarks for Author):

Few minor corrections/revisions are detailed below.

line 121-123 Olig2 positive cells could be oligodendrocyte precursors in addition to motor neuron precursors. Additionally, the CC1 antibody is typically used to label oligodendrocytes in tissue; in vitro the marker O4 would be preferred for all cells of the oligodendrocyte lineage, while A2B5 would be specific oligodendrocyte precursors and MBP or PLP for more mature myelinating cells.

>> The Reviewer is right: Olig2 is also a marker for oligodendrocyte precursors and we now mention this within the text. In addition, following the input of this Referee, we analyzed by Western Blot the presence of MBP in the total lysate of Healthy and ALS^{C9orf72} cultures (and murine spinal cord as positive control). We did not detect any signal in the hiPSC-derived cultures, confirming the absence of mature oligodendrocytes in our model (new Appendix Figure S1D).

lines 128-130 TEM morphological similarities of C9 to TBK1 mutant are informative, and the cells do look similar to each other, but it but difficult to assess the importance without a "normal" control as baseline for comparison.

>> As requested, a TEM image showing the absence of cytosolic aggregates in Healthy MN has been added.

some of the error bars in Fig 1G quantification seem to be right-shifted, as they present align between the c9 of one timepoint and the controls of the next time.

>> We apologize for this mistake; the alignment of the bars and stars with the graph of Fig 1G has been corrected.

Line 194- A rationale is required as to why TBK1 was chosen here. Since it replicates all of the effects, readers may question if this would be true of any/all ALS lines. Either another line that has negative results is needed, or a strong rationale is needed as to why TBK1 specifically would be anticipated to have a similar phenotype. Otherwise, it is difficult to determine if this finding is applicable to ALS as a whole, or potentially a subset of genetic categories.

>> The choice of using ALS^{TBK1} as additional line is justified by the strongly convergent autophagic phenotype displayed by these cells with ALS^{C9orf72} ones. Indeed, MN from both ALS genotypes accumulate aberrant aggregates (which sequester CBP), SQSTM1 aggregates, and show an early phase autophagy blockade. Moreover, our previous study (Catanese et al., 2019) showed already a certain degree of similarity in the pathological phenotypes displayed by these particular ALS-related MN. Based on these data, and on the notion that the two proteins contribute in controlling autophagy at the same stage of autophagosome development, we speculated that CREB dysregulation might represent a shared alteration as well. This is now stated within the manuscript (lines 198-201).

We are pleased to inform you that your manuscript is accepted for publication and is now being sent to our publisher to be included in the next available issue of EMBO Molecular Medicine.

Corresponding Author Name: Alberto Catanese

Manuscript Number: